# Improving Multi-Task Generalization via Regularizing Spurious Correlation

**Ziniu Hu**[1][*]**, Zhe Zhao**[2]**, Xinyang Yi**[2]**, Tiansheng Yao**[2]**, Lichan Hong**[2]**, Yizhou Sun**[1]**, Ed H. Chi**[2]
[1]University of California, Los Angeles, {`bull, yzsun`}@cs.ucla.edu
[2]Google Research, Brain Team, {`zhezhao,xinyang,tyao,lichan,edchi`}@google.com

## Abstract

Multi-Task Learning (MTL) is a powerful learning paradigm to improve generalization performance via knowledge sharing. However, existing studies find that MTL could sometimes hurt generalization, especially when two tasks are less correlated. One possible reason that hurts generalization is spurious correlation, i.e., some knowledge is spurious and not causally related to task labels, but the model could mistakenly utilize them and thus fail when such correlation changes. In MTL setup, there exist several unique challenges of spurious correlation. First, the risk of having non-causal knowledge is higher, as the shared MTL model needs to encode all knowledge from different tasks, and causal knowledge for one task could be potentially spurious to the other. Second, the confounder between task labels brings in a different type of spurious correlation to MTL. Given such label-label confounders, we theoretically and empirically show that MTL is prone to taking non-causal knowledge from other tasks. To solve this problem, we propose Multi-Task Causal Representation Learning (MT-CRL) framework. MT-CRL aims to represent multi-task knowledge via disentangled neural modules, and learn robust task-to-module routing graph weights via MTL-specific invariant regularization. Experiments show that MT-CRL could enhance MTL model's performance by 5.5% on average over Multi-MNIST, MovieLens, Taskonomy, CityScape, and NYUv2, and show it could indeed alleviate spurious correlation problem.

## 1 Introduction

Multi-Task Learning (MTL), a learning paradigm (Caruana, 1997; Zhang & Yang, 2018) aiming to train a single model for multiple tasks, is expected to benefit the overall generalization performance than single-task learning (Maurer et al., 2016; Tripuraneni et al., 2020) given the assumption that there exists some common knowledge to handle different tasks. However, recent studies observed that, when two tasks are less correlated, MTL could lead to even worse overall performance (Parisotto et al., 2016; Zhang et al., 2021). A line of works (Yu et al., 2020; Wang et al., 2021; Fifty et al., 2021) resort performance drop to optimization challenge because conflicting tasks might compete for model capacity. However, both Standley et al. (2020) and our analysis in Section 2.2 show that, even with an over-parameterized model that achieves low MTL training loss, the final generalization performance could be worse than single-task learning. This finding motivates us to think about the following question: Are there any intrinsic problems in MTL that hurt generalization?

One widely studied issue that influences generalization is the spurious correlation problem (Geirhos et al., 2019, 2020), i.e., correlation that only existed in training datasets due to unobserved confounders (Lopez-Paz, 2016), but not causally correct. For example, as Beery et al. (2018) discussed, when we train an image classification model to identify cows with a biased dataset where cows mostly

---

[*]This work was done when Ziniu was an intern at Google.

appear in pastures, the trained cow classification model could exploit the features of background (e.g., pastures) to make prediction. Thus, when we apply the classifier to another dataset where cows also appear in other locations such as farms or rivers, it will fail to generalize (Nagarajan et al., 2021).

When it comes to MTL setting, there exist several unique challenges to handle spurious correlation problem. **First, the risk of having non-causal features is higher**. Suppose each task has different sets of causal features. To train a single model for all these tasks, the shared representation should encode all required features. Consequently, the causal features for one task could be potentially spurious to the other tasks, and such risk could be even higher with an increasing number of tasks. **Second, the confounder that leads to spurious correlation is different**. Instead of the standard confounders between feature and label, the nature of MTL brings in a unique type of confounders between task labels, e.g., correlation between tasks' labels could change in different distributions. For example, when we train a MTL model to solve both cow classification and scene recognition tasks, its encoder needs to capture both foreground and background information, and the spurious correlation between the two tasks in training set could mislead per-task model to utilize irrelevant information, e.g., use background to predict cow. Given such label-label confounders that are unique for MTL, we theoretically prove that MTL is prone to taking non-causal knowledge learned from other tasks. We then conduct empirical analysis to validate the hypothesis. In summary, we point out the unique challenges of spurious correlation in MTL setup, and show that it indeed influences multi-task generalization.

In light of the analysis, we try to solve the spurious correlation problem in MTL. Among all the knowledge learned in the shared representation layer through end-to-end training, an ideal MTL framework should learn to leverage only the causal knowledge to solve each task by identifying the correct causal structure. Following the recent advances that enable causal learning in an end-to-end learning model (Schölkopf et al., 2021; Mitrovic et al., 2021), we propose a Multi-Task Causal Representation Learning (MT-CRL) framework, aiming to represent the multi-task knowledge via a set of disentangled neural modules instead of a single encoder, and learn the task-to-module causal relationship jointly. We adopt de-correlation and sparsity regularization over popular Mixture-of-Expert (MoE) architecture (Shazeer et al., 2017). The most critical and challenging step is to learn the causal graph in the MTL setup, which requires distinguishing the genuine causal correlation from spurious ones for all tasks. Motivated by the recent studies that invariance could lead to causality (Ahuja et al., 2020; Koyama & Yamaguchi, 2021), we propose to penalize the variance of gradients w.r.t. causal graph weights across different distributions. On a high level, this invariance regularization encourages the causal graph to assign higher weights to the modules that are consistently useful. In contrast, the modules encoding spurious knowledge that cannot consistently achieve graph optimality are assigned lower weights and be discarded by task predictors.

We evaluate our method on existing MTL benchmarks, including Multi-MNIST, MovieLens, Taskonomy, CityScape, and NYUv2. For each dataset, to mimic distribution shifts, we adopt some attribute information given in the dataset, such as the released time of the movie or district of a building, to split train/valid/test datasets. The results show that MT-CRL could consistently enhance the MTL model's performance by 5.5% on average, and outperform both the MTL optimization and robust machine learning baselines. We also conduct case studies to show that MT-CRL indeed alleviate spurious correlation problem in MTL setup.

The key contributions of this paper are as follows:

- We are the first to analyze spurious correlation problem in MTL setup, and point out several key challenges unique to MTL with theoretical and empirical analysis.
- We propose MT-CRL with MTL-specific invariant regularizers to elleviate spurious correlation problem, and enhances the performance on several MTL benchmarks.

## 2 Analyzing Spurious Correlation in MTL

To systematically analyze the spurious correlation problem in MTL, we first assume that data and task labels are generated by ground-truth causal mechanisms described in Suter et al. (2019). We denote $X$ as the variable of observed data, and each data is associated with $K$ latent generative factors $\mathbb{F} = \{F_i\}_{i=1}^K$ representing different semantics of the data (e.g., color, shape, background of an image). We follow Schölkopf et al. (2021) to assume that the data $X$ is generated by disentangled causal mechanisms $P(X|F_i)$, such that $P(X|\mathbb{F}) = \prod_{i=1}^K P(X|F_i)$.

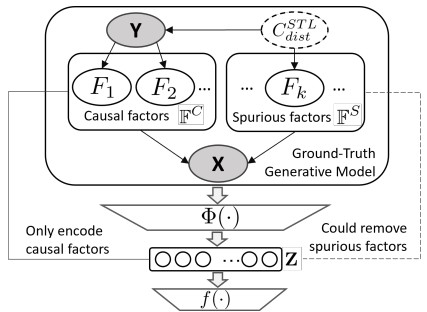

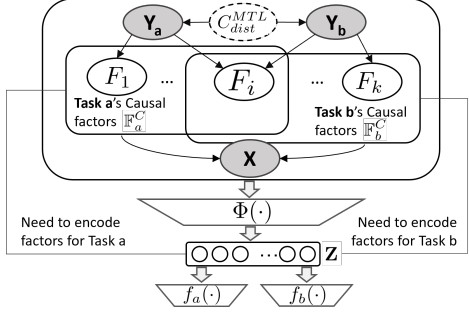

Figure 1: Spurious correlation in **Single-Task Learning** is mainly caused by factor-label confounders $C_{dist}^{STL}$. We could remove spurious factors $\mathbb{F}^S$ from representation $Z$.

Figure 2: Spurious correlation in **Multi-Task Learning** could be caused by label-label confounders $C_{dist}^{MTL}$. Factors for both tasks $\mathbb{F}_a^C$ and $\mathbb{F}_b^C$ need to be encoded and potentially spurious.

As $\mathbb{F}$ represents high-level knowledge of the data, we could naturally define task label variable $Y_t$ for task $t$ as the cause of a subset of generative factors. We denote $\mathbb{F}_t^C$ as a subset of causal feature variables within $\mathbb{F}$ that are causally related to each task variable $Y_t$, and we could define $\mathbb{F}_t^S = \mathbb{F} \setminus \mathbb{F}_t^C$ as a subset of non-causal feature variables to task $t$, such that $P(Y_t|\mathbb{F}) = P(Y_t|\mathbb{F}_t^C)$. In other words, changing the values of any non-causal factors in $\mathbb{F}_t^S$ does not change the conditional distribution.

Note that the discussion so far is based on the assumption that the ground-truth causal generative model is known. In a real-world learning setting, however, we are only given a supervised dataset $(X, Y)$ without access to generative factors $\mathbb{F}$. To solve the task, a neural encoder $\Phi(\cdot)$ is required to extract representation $\mathbf{Z}$ from the data that encodes the information about the causal factors, on top of which a task predictor $f(\cdot)$ could predict the label.

## 2.1 Spurious Correlation Problem

Based on the ground-truth generative model, an ideal predictor for each task should only utilize the causal factors, and keep invariant to any intervention on non-causal factors. However, in real-world problems, it is hard to achieve an invariant predictor due to the spurious correlation issue due to unobserved confounders $C_{dist}$ (Lopez-Paz, 2016). Formally, confounders are variables that influence the two connected variables' correlation, and such correlation could change under different distribution (different value of $C_{dist}$), thus the model exploiting such spurious correlation will fail to generalize. Below we summarize the differences of spurious correlation problems for single-task and multi-task learning settings:

**Single-Task Learning (STL).** As illustrated in Figure 1, the label-factor confounders for single task learning $C_{dist}^{STL}$ connects non-causal factors $F \in \mathbb{F}^S$ and task label $Y$, bringing in spurious correlation. For example, temperature could confound crime and ice cream consumption. When the weather is hot, both crime rates and ice cream sales increase, but these two phenomena are not causally related. Based on the proof by Nagarajan et al. (2021); Khani & Liang (2021), such spurious correlation could lead the model to use non-causal factors, and thus hurt generalization performance.

**Multi-Task Learning (MTL).** In the MTL setting, there exist several unique challenges to handle spurious correlation. First, the risk of having non-causal features is higher. As is illustrated in Figure 2, the shared encoder $\Phi$ needs to encode all the factors causally related to each task in the representation $Z$. Therefore, for each task, all non-overlapping factors from other tasks could be potentially spurious. Second, besides the standard label-factor confounders $C_{dist}^{STL}$ for each single task introduced above, we define label-label confounders $C_{dist}^{MTL}$ connecting multiple tasks' label $\{Y\}$. Such confounder is unique to MTL setting.

As an example, consider two binary classification tasks, with $Y_a$ and $Y_b$ as variables from $\{\pm 1\}$ for task label. The two labels' correlation $P(Y_a = Y_b) = m_C$ could change with different confounder $C_{dist}^{MTL} = C$. We assume the two tasks have non-overlapping factors $F_a$ and $F_b$ drawn from Gaussian distribution. We then show MTL model with both two factors as input will utilize non-causal factors:

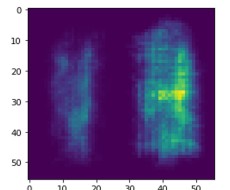
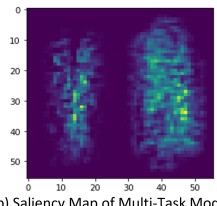

(a) Saliency Map of Single-Task Model  (b) Saliency Map of Multi-Task Model

|  | Multi-SEM | | Multi-MNIST | |
|---|---|---|---|---|
|  | STL | MTL | STL | MTL |
| $\text{Acc}_{train}$ | 0.931 | 0.936 | 0.981 | 0.987 |
| $\text{Acc}_{val}$ | 0.906 | 0.882 | 0.874 | 0.846 |
| $\rho_{spur}$ | 0.128 | 0.246 | 0.261 | 0.328 |

Figure 3: The gradient saliency map of right-side digit classifier. The model trained by MTL exploits left pixels (spurious) more.

Table 1: Empirical results of multi-task (MTL) and single-task learning (STL) model on synthetic datasets with changing $C_{dist}^{MTL}$.

**Proposition 1** *Given $m_C \neq 0.5$, the Bayes Optimal per-task classifier has non-zero weights to non-causal factor. Given $m_C = 0.5$ and limited training dataset, the trained per-task classifier will assign non-zero weights to non-causal factor as noise.*

Detailed proof is in Appendix A. Therefore, in this linear classification example, when we deploy the trained model to a new distribution with changed label-label confounder $C_{dist}^{MTL}$, the model trained by MTL that utilizes non-causal factors generalize relatively worse. On the contrary, the model trained by STL don't need to encode all causal factors from two tasks. Assuming there is no task-label confounder $C_{dist}^{STL}$ in each task's dataset, the trained model could remove non-causal factors from representation.

## 2.2 Empirical Experiments

In the following, we conduct experiments to validate the claims. As there is no existing MTL datasets specifically designed to analyze spurious correlation problem, we construct synthetic Multi-SEM (Rosenfeld et al., 2021) and Multi-MNIST (Harper & Konstan, 2016) datasets with known causal structure to study whether the model trained by MTL indeed exploits more non-causal factors, and how the spurious correlation influences multi-task generalization. Dataset details are in Appendix D.1.

**Spurious Score.** As we know the ground-truth causal structure for the two datasets, we could quantify how much a model utilizes the non-causal factors. Following the gradient saliency map proposed by Simonyan et al. (2014), we calculate the average absolute gradients w.r.t each factor as $Grad(F) = \sum_{(x(\mathbb{F}),y)\in D} \left| \frac{\partial\left(f(\Phi(x))[y]\right)}{\partial F} \right|$, which measures how much a model leverage this factor to make prediction. We then define the spurious score $\rho_{spur}$ as the proportion of average gradients over non-causal feature $\rho_{spur} = \frac{\sum_{F\in\mathbb{F}^S} Grad(F)}{\sum_{F\in\mathbb{F}} Grad(F)}$.

**Empirical Results.** We train a shared-bottom model via Multi-task learning (MTL) and single-task learning (STL) over the two datasets and report both the training and test accuracy with spurious ratio $\rho_{spur}$ in Table 1. As illustrated, the test accuracies of MTL for both Multi-SEM and Multi-MNIST datasets are both worse than STL. The training accuracies of MTL are very similar to STL, meaning that the performance drop is not due to the optimization difficulty that many previous works try to address. The spurious ratio $\rho_{spur}$ of MTL is much higher than the STL, which means that it exploits more non-causal factors. To give a more straightforward illustration, we plot the gradient saliency map of the right-side digit classifier for Multi-MNIST in Figure 3. The model trained by MTL utilizes more left-side pixels, which are non-causal to the final prediction. We also show the results of Multi-SEM with more than 2 tasks in Appendix B. These findings support our hypothesis that with spurious correlation caused by label-label confounder $C_{dist}^{MTL}$, models trained by MTL is more prone to leverage non-causal knowledge than STL, and thus influence generalization performance.

## 3 Method

Based on the previous analysis of the spurious correlation problem in MTL, we now introduce a Multi-Task Causal Representation Learning (MT-CRL) framework with the goal that the per-task predictor only leverages the causal knowledge instead of spurious correlation. The high-level idea

of the framework is to reconstruct the ground-truth causal mechanisms introduced in section 2 through end-to-end representation learning. To accomplish this goal, the framework aims to 1) model multi-task knowledge via a set of disentangled neural modules; 2) learn the task-to-module causal graph that is optimal across different distributions. With the correct causal graph as routing layer, per-task predictor only utilizes outputs from causally-related modules, thus alleviating the spurious correlation problem. We introduce the two crucial designs as follows.

## 3.1 Modelling via Disentangled Neural Modules

In order to alleviate spurious correlation, an ideal MTL model should learn the multi-task knowledge in the shared representation while identifying which part of the knowledge is causally related to each task. However, directly conducting causal discovery is impossible if all the knowledge is fused in a single shared encoder. Thus, we seek to adopt a modularized architecture in which each module encodes disentangled knowledge, and thus enable modeling causal relationship between task and modules. We adopt Multi-gate Mixture-of-Experts (MMoE) (Ma et al., 2018), a variant of MoE (Shazeer et al., 2017) architecture tailored for MTL setting, as our underlying model. Specifically, we have $K$ different neural modules as shared encoders $\Phi = \big[\Phi_i(\cdot)\big]_{i=1}^{K}$. Given a batch of input data $\mathbf{X} = \{x_n\}_{n=1}^{B}$ with batch size $B$, we extract $k$ representations via different neural modules, i.e., $\mathbf{Z}_i = \Phi_i(X) \in \mathbb{R}^{B \times d}$. Based on sparsity assumption of the causal mechanisms Parascandolo et al. (2018); Bengio et al. (2020); Lachapelle et al. (2021), only a few modules should be causally related to each task. Therefore, on top of the learned neural modules, we learn a task-to-module routing graph, aiming to estimate which module is causally related to each task. We model the bipartite adjacency (a.k.a. bi-adjacency) matrix $A = \text{sigmoid}(\theta) \in [0,1]^{T \times K}$ by applying sigmoid over a learnable parameter $\theta$ to enforce the range constraint. Note that original MMoE adopts softmax to get gate vector, which encourages only a small portion of modules being utilized for each task. Our graph modelling allows multiple modules utilized for each task. With the correct graph weights $A$ as routing layer, we could utilize only the causally related modules and make predictions with per-task predictor $f_t(\cdot)$ as $\hat{Y}_t(\mathbf{X}) = f_t\big(\sum_i A_{t,i} \cdot \Phi_i(\mathbf{X})\big)$.

**Disentangling Modules.** One of the main properties of the causal mechanisms we introduced in section 2 is disentanglement, such that each factor represents a different view of the data, and changing the value of one factor does not influence the others. If without explicit constraints, the learned modules' outputs could still be correlated and hinder the causal structure learning. Therefore, we need to add regularization to disentangle these modules during training.

Most existing disentangled representation learning methods are under the generative modeling framework, e.g. VAE (Higgins et al., 2017) or GAN (Chen et al., 2016). However, Locatello et al. (2019) argues that without explicit supervision, it is hard for generative models to learn correct disentangled factors. We therefore only borrow the regularization methods utilized in existing generative disentangled representation works (Cheung et al., 2015; Cogswell et al., 2016) to directly penalize the correlation of learned modules. Specifically, we regularize the in-batch Pearson correlation $\rho(\mathbf{Z}_i, \mathbf{Z}_j)$ between every pair of output dimensions from different representation matrices $\mathbf{Z}_i$ and $\mathbf{Z}_j$, as:

$$\rho(\mathbf{Z}_i, \mathbf{Z}_j) = \frac{Cov(\mathbf{Z}_i, \mathbf{Z}_j)}{\sqrt{Cov(\mathbf{Z}_i, \mathbf{Z}_i)}\sqrt{Cov(\mathbf{Z}_j, \mathbf{Z}_j)}}, \text{ where } Cov(\mathbf{Z}_i, \mathbf{Z}_j) = \big[\mathbf{Z}_i - \overline{\mathbf{Z}_i}\big]^T \big[\mathbf{Z}_j - \overline{\mathbf{Z}_j}\big]. \quad (1)$$

By minimizing the Frobenius norm of the correlation matrix $\rho$ for every two different representation pairs, we could enforce the encoder $\Phi$ to extract disentangled representations.

$$\mathcal{L}_{decor}(\Phi) = \lambda_{decor} \cdot \sum_{i=1}^{k} \sum_{j=i+1}^{k} \Big\| \rho\big(\Phi_i(X), \Phi_j(X)\big) \Big\|_F^2. \quad (2)$$

**Task-to-Module Graph Regularization.** Based on sparsity assumption of the causal mechanisms (Parascandolo et al., 2018; Bengio et al., 2020; Lachapelle et al., 2021), each task is causally related to only a few modules. To learn the graph structure, existing works (Zheng et al., 2018; Ng et al., 2019; Lachapelle et al., 2020) propose to to fit structural equation model (SEM) with sparsity regularization over the graph weights. We adopt a similar sparse regularization with an entropy balancing term (Hainmueller, 2012) over the bi-adjacency matrix $A$ weights of the task-to-module

routing graph:

$$\mathcal{L}_{graph}(A) = \lambda_{sps} \cdot ||A||_1 - \lambda_{bal} \cdot \text{Entropy}\left(\frac{\sum_t A_{t,*}}{\sum_{t,i} A_{t,i}}\right). \tag{3}$$

Note that the entropy term aims at keeping the causal weights for each module $i$ summing over all the tasks to be balanced. This could help avoid degenerate solutions in which only a few modules are utilized. By combining the two regularizations in Eq.(2) and Eq.(3) with per-task supervised risk term $R_t(\Phi, A_t, f_t) = \sum_{(\mathbf{X}, \mathbf{Y}_t) \in \mathcal{D}} L_t(\hat{Y}_t(\mathbf{X}), \mathbf{Y}_t)$, we get the regularized loss as:

$$\tilde{\mathcal{L}}(\Phi, A, f) = \sum_{t \in \mathcal{T}} R_t(\Phi, A_t, f_t) + \mathcal{L}_{decor}(\Phi) + \mathcal{L}_{graph}(A). \tag{4}$$

### 3.2 Causal Learning via Graph-Invariant Regularization

It is critical and challenging to learn the correct causal graph, which requires distinguishing the true causal correlation from spurious ones. Motivated by the recent studies of robust machine learning that a predictor invariant to multiple distributions could learn causal correlation (Ahuja et al., 2020; Koyama & Yamaguchi, 2021), we assume the true causal relationship to be optimal across different distributions. To do so, we assume to have access to multiple slices of datasets collected from different environments $e \in \mathcal{E}$ in which the confounder $C_{dist}^{MTL}$ that controls task correlation might change. For example, one natural choice is to consider train/valid dataset split (the setting we utilize in experiment), or assume the training set is split into multiple slices with different attributes. We desire the task-to-module graph weights $A$ and per-task predictor $f_t$ to be optimal across all environments $e \in \mathcal{E}$. Formally, we aim to solve the following bi-level optimization problem:

$$\min_{\Phi, A, f} \tilde{\mathcal{L}}(\Phi, A, f) \quad \text{s.t.} \quad A_t, f_t \in \arg\min_{A, f} R_t^e(\Phi, A, f), \forall\, t \in \mathcal{T}, e \in \mathcal{E}. \tag{5}$$

where $R_t^e$ denotes the risk over data slice in environment $e$. This optimization problem could be regarded as a multi-task version of IRM. Based on Theorem 9 described in Ahuja et al. (2020), by enforcing invariance over a sufficient number of environments that exhibit distribution shifts (i.e., changes of confounder $C_{dist}^{MTL}$), per-task predictors should only utilize modules that are consistently helpful to the task, and assign zero weights to modules that encode non-causal factors to the task, and thus alleviate spurious correlation and help out-of-distribution generalization. Even if all data are sampled from the same distribution and there are no distribution shifts, invariance could also help eliminate noisy correlation due to the limited training dataset and help in-distribution generalization.

**Invariant Optimality of Task-to-Module Graph for MTL.** As discussed in IRM, the bi-leveled optimization problem in Eq.(5) is highly intractable, especially with complex and non-linear $\Phi$. To implement a practical optimization objective, IRM proposes to softly regularize the gradient of the task-predictor at different environments to enforce it to be optimal:

$$\min_{\Phi, A, f} \left( \tilde{\mathcal{L}}(\Phi, A, f) + \sum_{t \in \mathcal{T}} \sum_{e \in \mathcal{E}} \left\| \nabla_{A=A_t, f=f_t} R_t^e(\Phi, A, f) \right\|^2 \right). \tag{6}$$

However, as is discussed in IRM paper, if the complexity of a task-predictor $f$ is much larger than the number of environments, it could learn an over-fitted solution that makes gradient zero but does not achieve invariance. IRM adopts a fixed all-one vector as predictor to reduce complexity. This approach **is not applicable to MTL setup**, as the optimal task-predictors $f_t^*$ for different task $t$ could be very distinctive and complex, and we cannot use a fixed uniform predictor for all tasks.

To strike a balance between invariance and complexity of multi-task predictors, we propose only to regularize the gradient of the task-to-module routing graph while assuming the complex predictor $f_t$ for each task is fixed at each iteration. We call this modification as **Graph-Invariant Risk Minimization (G-IRM)**, which is designed specifically to MTL setup:

$$\min_{\Phi, A, f} \left( \tilde{\mathcal{L}}(\Phi, A, f) + \lambda_{G\text{-}IRM} \cdot \mathcal{L}_{G\text{-}IRM}(\Phi, A | f) \right). \tag{7}$$

By adopting the similar gradient penalty term as adopted in IRM, we define $\mathcal{L}_{G\text{-}IRM}^{Norm}$ as:

$$\mathcal{L}_{G\text{-}IRM}^{Norm}(\Phi, A | f) = \sum_{t \in \mathcal{T}} \sum_{e \in \mathcal{E}} \left\| \nabla_{A=A_t} R_t^e(\Phi, A, f_t) \right\|^2. \tag{8}$$

As we assume $f_t$ is fixed for invariance regularization term $\mathcal{L}_{G\text{-}IRM}^{Norm}$, we only calculate gradient and optimize for $\Phi$ and $A$, but not updating $f_t$. This could avoid the over-parametrized predictor $f_t$ finding a trivial solution to achieve zero gradients instead of learning the correct causal correlation. Similar trick is utilized in (Ahmed et al., 2021). Note that the gradient w.r.t each graph weight means whether a module could help reduce the risk for this task. Therefore, by penalizing the invariance regularization, the modules containing non-causal factors will be assigned zero weights.

In the experiments, we observe that at the early optimization stage, the model has non-zero gradients for all parameters, including the graph weights, thus directly regularizing the gradient norm might influence the optimization. Therefore, we propose a modified version of gradient regularization $\mathcal{L}_{G\text{-}IRM}^{Var}$ that penalizes the variance of the task-to-module graph's gradient on different environments:

$$\mathcal{L}_{G\text{-}IRM}^{Var}(\Phi, A|f) = \sum_{t \in \mathcal{T}} \sum_{e \in \mathcal{E}} \frac{1}{|\mathcal{E}|} \left\| \nabla_{A=A_t} R_t^e(\Phi, A, f_t) - \text{Avg}_e\left(\nabla_{A=A_t} R_t^e\right) \right\|^2. \tag{9}$$

By minimizing $\mathcal{L}_{G\text{-}IRM}^{Var}$, we force all the learned modules to have similar gradients across different environments, and not overfit only to some of the environments. It still allows some modules to have non-zero gradients as long as it's the same across environments, and relies on loss term $\tilde{\mathcal{L}}$ to update these weights, while $\mathcal{L}_{G\text{-}IRM}^{Norm}$ forces all gradient to be zero. Therefore, $\mathcal{L}_{G\text{-}IRM}^{Var}$ is a loose regularization that not influences the overall optimization too much. It shares similar intuition of REx (Krueger et al., 2021) that penalizes risk variance, while $\mathcal{L}_{G\text{-}IRM}^{Var}$ penalize gradient variance. We provide pseudo-code of MT-CRL framework in Appendix C.

## 4 Experiment

In this section, we evaluate whether MT-CRL could benefit the performance of MTL models on existing benchmark datasets, and study whether it could indeed alleviate spurious correlation.

**Experimental Setup.** One key ingredient of our MT-CRL is to achieve the optimality of causal graph over different distributions. However, we might not access multiple environmental labels in most real-world multi-task learning datasets. Therefore, we adopt a more realistic setup, such that we only assume to have a single validation set that contains unknown distribution shifts (i.e. change of confounder $C_{dist}^{MTL}$) compared to the training dataset. We thus could utilize training and valid sets as two environments to calculate invariance regularization, while we only utilize the training set to calculate task loss to avoid the task predictor overfits. Note that in this way, our method could get access to the label information in the validation set. To avoid the possibility that the performance improvement is brought by additional label, for all the other baseline methods, we also add the validation data into the training set to calculate task loss and learn MTL model.

**Dataset.** We choose five widely-used real-world MTL benchmark datasets, i.e., Multi-MNIST (Sun, 2019), MovieLens (Harper & Konstan, 2016), Tasknomy (Zamir et al., 2018), NYUv2 (Silberman et al., 2012) and CityScape (Cordts et al., 2016), and try to determine train/valid/test split such that there exist distribution shifts between these sets. Dataset details are in Appendix D.2. Note that except NYUv2, our data split is the same as the default split settings of these datasets, which also try to test model's capacity to generalize across domains.

**Baselines.** As MT-CRL is a regularization framework built upon modular MTL architecture (in this paper we choose MMoE as instantiation, but it can be applied to other modular networks), we mainly compare with two gradient-based multi-task optimization baselines: **PCGrad** (Yu et al., 2020) and **GradVac** (Wang et al., 2021). We also compare with two domain generalization baselines: **IRM** (Ahuja et al., 2020) and **DANN** (Ganin et al., 2016). For IRM we adopt different per-task predictors instead of all-one vector to adapt MTL setup, and calculate penalty via Eq. (6).

**Hyper-Parameter Selection.** For a fair comparison, all methods are based on the same MMoE architecture. Our methods contain a lot of hyper-parameters, including some model specific ones such as number of modules ($K$) and regularization specific ones. To avoid the case that performance improvement is caused by extensive hyper-parameter tuning, we mainly search optimal model hyper-parameter on Vanilla MTL setting, and use for all baselines. For regularization specific parameters, we take Multi-MNIST, the simplest dataset among the testbed, to find a optimal combination, and use for all other datasets. Detailed selection procedure and results are shown in Appendix H.

| Methods | Multi-MNIST | MovieLens | Taskonomy | CityScape | NYUv2 | Avg. |
|---|---|---|---|---|---|---|
| Vanilla MTL | (—baseline to calculate relative improvement—) | | | | | |
| Single-Task Learning | +3.3% | +0.2% | -2.5% | -2.4% | -12.2% | -2.7% |
| MTL + PCGrad | +4.5% | +0.2% | +3.1% | +2.1% | +7.4% | +3.5% |
| MTL + GradVac | +4.6% | +0.3% | +3.5% | +2.1% | +7.2% | +3.5% |
| MTL + DANN | +4.1% | +0.4% | +1.2% | +0.3% | -0.4% | +1.1% |
| MTL + IRM | +5.0% | +0.4% | +1.1% | +0.6% | -0.1% | +1.4% |
| MT-CRL w/o $\mathcal{L}_{G\text{-}IRM}$ | +5.9% | +0.2% | +3.2% | +1.5% | +4.3% | +3.0% |
| MT-CRL with $\mathcal{L}_{G\text{-}IRM}^{Norm}$ | +7.8% | +1.0% | +6.5% | **+2.9%** | +8.0% | +5.2% |
| MT-CRL with $\mathcal{L}_{G\text{-}IRM}^{Var}$ | **+8.1%** | **+1.1%** | **+7.1%** | +2.8% | **+8.2%** | **+5.5%** |

Table 2: Relative Performance improvement of different multi-task learning (MTL) strategies compared to vanilla MTL baseline. Detailed results for each task are shown in Table 6-10 in Appendix E.

| Disentangled Reg. | | Graph Reg. | | Multi-MNIST |
|---|---|---|---|---|
| $\mathcal{L}_{decor}$ | $\mathcal{L}_{\beta\text{-VAE}}$ | $\mathcal{L}_{sps}$ | $\mathcal{L}_{bal}$ | Accuracy |
| ✓ | ✗ | ✓ | ✓ | 0.915 ± 0.018 |
| ✗ | ✓ | ✓ | ✓ | 0.896 ± 0.024 |
| ✗ | ✗ | ✓ | ✓ | 0.882 ± 0.020 |
| ✓ | ✗ | ✗ | ✗ | 0.891 ± 0.016 |
| ✓ | ✗ | ✗ | ✓ | 0.903 ± 0.017 |
| ✓ | ✗ | ✓ | ✗ | 0.908 ± 0.021 |

Table 3: **Ablation Studies** of disentangled and Graph regularization components in MT-CRL, evaluated on Multi-MNIST dataset.

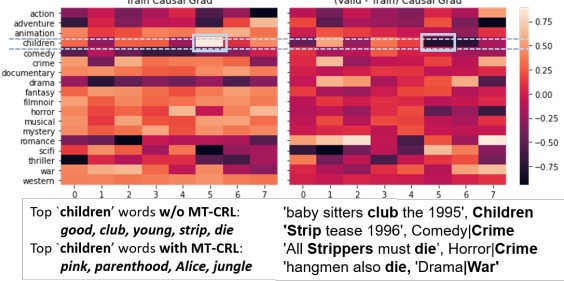

Figure 4: Task-to-Module gradients of model without MT-CRL show Module 5 is spurious. MT-CRL could help alleviate spurious correlation.

## 4.1 Experiment Results

As each task has a different evaluation metric and cannot be directly compared, we calculate the relative performance improvement of each method compared to vanilla MTL, and then average the relative improvement for all tasks of each dataset. As summarized in Table 2, the average improvement of MT-CRL with $\mathcal{L}_{G\text{-}IRM}^{Var}$ is 5.5%, significantly higher than all other baseline methods. The most critical step of MT-CRL is to learn correct causal graph. We therefore report MT-CRL with different invariance regularization. As is shown in the last block, $\mathcal{L}_{G\text{-}IRM}^{Var}$ achieve better results for most datasets than $\mathcal{L}_{G\text{-}IRM}^{Norm}$, while removing the invariance regularization could significantly drop the relative performance. Compared to IRM which calculate gradient and update per-task predictors, MT-CRL uses disentangled modules and G-IRM to avoid overfitting to achieve invariance. Results show that for datasets with large amount of tasks, e.g., Taskonomy and NYUv2, MT-CRL significantly outperform IRM, showing the modification is more suitable for MTL setup.

**Ablation Studies.** We then study the effectiveness of the other two components in MT-CRL, i.e., disentangled and graph regularization. We mainly report the ablation studies on Multi-MNIST in table 3 as it's relatively small so that we could quickly get the results of all combinations.

For disentangled regularization, after removing $\mathcal{L}_{decor}$, the performance drops from 0.915 to 0.882, which fits our discussion that we cannot conduct causal learning over entangled modules. We also explore one classical generative disentangled representation method, i.e., $\beta$-VAE. As shown in the table, the results of using $\beta$-VAE are 0.896, lower than our utilized decorrelation regularization.. We hypothesize that this is probably because not all generative factors are useful for downstream tasks. Generative objectives might compete for the model capacity and in addition, the unused factors could be potentially spurious.

Another key component is graph regularization. After removing both $\mathcal{L}_{sps}$ and $\mathcal{L}_{bal}$, the performance drops to 0.891. This show that even if invariance regularization could penalize non-causal modules, it would be better to force their weights to be zero via sparsity regularization, and to be non-degenerate

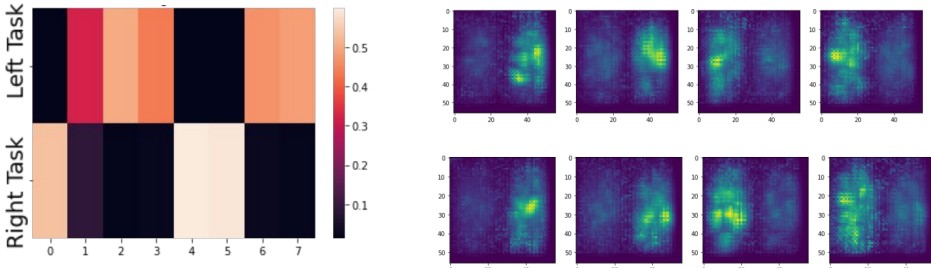

Figure 5: (valid-train) Task-to-Module gradients of model **with** MT-CRL on Multi-MNIST.

via balance regularization. We also conduct ablation studies to remove either $\mathcal{L}_{sps}$ or $\mathcal{L}_{bal}$, and results show both are important, and combining the two could help to achieve the best results.

**Case Study.**    To show that real-world MTL problem indeed have spurious correlation problem and our MT-CRL could alleciate it, we take MovieLens as an example to conduct case study. Each task is for different movie types, and bag-of-word of movie title is one of the features. We calculate the task-to-module gradients $\frac{\partial(f(\Phi(x))[y])}{\partial F}$ of the vanilla MMoE model without MT-CRL. We then visualize 'train' gradients, which shows how much each module is utilized to fit the training set, and 'valid-train' gradients, which shows how generalizable each module is. We find that module 5 is utilized for **children** movie, but harmful in valid set, indicating it is a spurious feature. We then use Grad-CAM to show that top words of module 5 include *strip* and *die*, which is not relevant to **children** movies. One possible reason is that some children movies contain the words *club*, which is often co-occurred with *strip* and *die* in **crime** and **war** movies. After adding our MT-CRL, the module assigned to 'children' movie attends *Pink*, *Parenthood*, *Alice* and *Jungle*.

We then show the (valid-train) Task-to-Module gradients over Multi-MNIST datasets. With MT-CRL, in Figure 14, each module's saliency map only focus on one side of pixels. By looking at each task output's saliency map, which help model to focus only on causal part, compared with Figure 3(b) that have high weights on both. We also show the detailed gradient saliency map and induced task similarity graph of MovieLens, Taskonomy in Appendix G. All these case studies show MT-CRL could indeed alleviate spurious correlation in real MTL problems.

## 5  Related Work

**Multi-Task Generalization.**    A deep neural model often requires a large number of training samples to generalize well (Arora et al., 2019; Cao & Gu, 2019). To alleviate the sample sparsity problem, MTL could leverage more labeled data from multiple tasks (Zhang & Yang, 2018). Most works studying multi-task generalization are based on a core assumption that the tasks are correlated. Earlier research directly define the task relatedness with statistical assumption (Baxter, 2000; Ben-David & Borbely, 2008; Lampinen & Ganguli, 2019). With the increasing focus on deep learning models, recent research decompose ground-truth MTL models into a shared representation and different task-specific layers from a hypothesis family (Maurer et al., 2016). With such decomposition, Tripuraneni et al. (2020) and Du et al. (2021) prove that a diverse set of tasks could help learn more generalizable representation. Wu et al. (2020) study how covariate shifts influence MTL generalization. Despite these findings, the core assumption of task relatedness might not be satisfied in many real-world applications (Parisotto et al., 2016; Zhang et al., 2021), in which tasks could even conflict with each other to compete model capacity, and the generalization performance of MTL could be worse than single-task training.

To solve the task conflict problem, a number of MTL model architectures have utilized modular (Misra et al., 2016; Lu et al., 2017; Rosenbaum et al., 2018; Ma et al., 2018; Guo et al., 2020) or attention-based (Liu et al., 2019; Maninis et al., 2019) design to enlarge model capacity while preserving information sharing. Our work is model-agnostic and could be applied to existing architectures to further solve the spurious feature problem. Another line of research alleviate task conflict during optimization. Some propose to balance the task weight via uncertainty estimation (Kendall et al., 2018), gradient norm (Chen et al., 2018), convergence rate (Liu et al., 2019), or pareto optimality (Sener & Koltun, 2018). Others directly modulate task gradients via dropping part of the conflict

gradient (Chen et al., 2020) or project task's gradient onto other tasks' gradient surface (Yu et al., 2020; Wang et al., 2021). Though these works successfully facilitate MTL model to converge easier, our analysis show that with spurious correlation, the MTL model with low training loss could still generalize bad. Therefore, our proposed MT-CRL that alleviates spurious correlation is orthogonal to these prior works, and could be combined to further improve overall performance.

**Spurious Correlation Problem.**  Due to the selection bias (Torralba & Efros, 2011; Gururangan et al., 2018) or unobserved confounding factors (Lopez-Paz, 2016), training datasets always contain spurious correlations between non-causal features and task labels, with which trained models often leverage non-causal knowledge and may fail to generalize Out-Of-Distribution (OOD) when such correlation changes (Nagarajan et al., 2021). To solve the spurious correlation problem, some fairness research pre-define a set of non-causal features (e.g., gender and underrepresented identity) and then explicitly remove them from the learned representation (Zemel et al., 2013; Ganin et al., 2016; Wang et al., 2019). Another line of robust machine learning research does not assume to know spurious features, but regularize the model to perform equally well under different distribution. Distributionally Robust Optimization (DRO) optimizes worst-case risk (Sagawa et al., 2020). Invariant Causal Prediction (ICP) learns causal relations via invariance testing (Peters et al., 2016). Invariant Risk Minimization (IRM) forces the final predictor to be optimal across different domains (Arjovsky et al., 2019). Risk Extrapolation (REx) directly penalizes the variance of training risk in different domains (Krueger et al., 2021). Another line of work aim at learning causal representation (Schölkopf et al., 2021), i.e., high-level variables representing different aspect of knowledge from raw data input. Most of these works try to recover disentangled causal generative mechanisms (Parascandolo et al., 2018; Bengio et al., 2020; Liu et al., 2020; Mitrovic et al., 2021). Despite the extensive study of spurious correlation in single-task setting, few work discuss it for MTL models. This paper is the first to point out the unique challenges of spurious correlation in MTL setup.

## 6   Conclusion

In this paper, we study spurious correlation problem in the Multi-Task Learning (MTL) setting. We theoretically and experimentally shows that task correlation can introduce special type of spurious correlation in MTL, and the model trained by MTL is more prone to leverage non-causal knowledge from other tasks than single-task learning. To solve the problem, we propose Multi-Task Causal Representation Learning (MT-CRL) which consists of: 1) a decorrelation regularizer to learn disentangled modules; 2) a graph regularizer to learn sparse and non-degenerate task-to-module graph; 3) G-IRM invariant regularizer. We show MT-CRL could improve performance of MTL models on benchmark datasets and could alleviate spurious correlation.

**Limitation Statement.**  Our analysis is based on label-label confounders. However, existing MTL datasets don't provide exact confounder changes to study spurious correlation problem. As mitigation, in analysis part, we create two synthetic datasets, and in experiment part, we adopt train/valid/test split with several attribution differences to mimic confounder changes. To further study spurious correlation in MTL, in the future, we'd like to construct benchmark MTL datasets with known confounder changes (or analyze how some key attribute changes lead to spurious correlation problem), build mathematical model based on it, and also explore and visualize which part of knowledge in real-world MTL datasets (e.g. Taskonomy) could be spuriously correlated to other tasks.

**Acknowledgement.**  We sincerely thanks anoymoused NeurIPS reviewers for their constructive comments and suggestions to improve this paper. We thank Huan Gui, Kang Lee, Alexander D'Amour, Xuezhi Wang, Jilin Chen and Minmin Chen for insightful discussion and suggestion for this work. We also thank Ang Li for technical support for running experiments on Taskonomy, and Thanh Vu for running CityScape and NYUv2 experiments.

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
