# A  Proof of Proposition 1

## A.1  Problem Definition

We consider two binary classification tasks, with $Y_a$ and $Y_b$ as variables from $\{\pm 1\}$ for task label. The task labels are drawn from two different probabilities. For simplicity, we assume the probability to sample the two label value is balanced, i.e., $P(Y = 1) = P(Y = -1) = 0.5$. Our conclusion could be extended to unbalanced distribution.

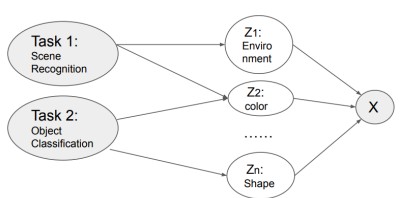

In this paper, we mainly study the spurious correlation between task labels. For simplicity, we define $P(Y_a = Y_b) = m_C, P(Y_a \neq Y_b) = 1 - m_C$, where $m_C$ denotes that this correlation could change by different confounder $C_{dist}^{MTL}$. In some environments $m_C \neq 0.5$, meaning that the two tasks are correlated in these environments. To sum up, we could define the probability table as:

Figure 6: Illustrative Diagram of Causal Generative Model in MTL setting

|           | $Y_a = 1$ | $Y_a = 0$ |
|-----------|-----------|-----------|
| $Y_b = 1$ | $m_C$     | $1 - m_C$ |
| $Y_b = 0$ | $1 - m_C$ | $m_C$     |

Table 4: Probability table for $P(Y_a, Y_b)$, where $m_C$ denotes the correlation between the two task label.

We consider two $d$-dimensional factors $F_a$ and $F_b$ representing the knowledge to tackle the two tasks. Both are drawn from Gaussian distribution:

$$F_a \sim \mathcal{N}(Y_a \cdot \mu_a, \sigma_a^2 I), \quad F_b \sim \mathcal{N}(Y_b \cdot \mu_b, \sigma_b^2 I) \tag{10}$$

with $\mu_a, \mu_b \in \mathbb{R}^d$ denote the mean vectors and $\sigma_a, \sigma_b$ are covariance vectors.

Our goal to learn two linear models $P(Y_{\{a/b\}}|F_a, F_b) = \text{sigmoid}(\beta F) = \text{sigmoid}(\beta_a F_a + \beta_b F_b)$. We first consider the setting that we're given infinite samples. If we assume there's no traditional factor-label spurious correlation in single task learning, the bayes optimal classifier will only take each task's causal factor as feature, and assign zero weights to non-causal factors. The factor with the regression vector $\beta_a = \frac{\mu_a}{\sigma_a^2}$ for bayes optimal classifier of task $a$ and $\beta_b = 2\frac{\mu_b}{\sigma_b^2}$ for bayes optimal classifier of task $b$.

## A.2  Bayes Optimal Classifier for Multiple-Task

When we train a single model using both tasks, the optimal Bayes classifier will utilize the other non-causal factor due to the influence of spurious correlation quantified by $m_C$. To prove it, we take the first task with label $Y_a$ as an example and derive the optimal Bayes classifier as:

$$P(Y_a|F_a, F_b) = \frac{P(Y_a, F_a, F_b)}{P(F_a, F_b)} = \frac{P(Y_a, F_a, F_b)}{\sum_{Y_a \in \{-1,1\}} P(Y_a, F_a, F_b)} \tag{11}$$

while the probability of $P(Y_a, F_a, F_b)$ could be written as:

$$P(Y_a, F_a, F_b) = P(Y_a, F_a) \cdot P(F_b|Y_a, F_a) \tag{12}$$

$$= P(Y_a, F_a) \cdot P(F_b|Y_a) \tag{13}$$

$$= P(Y_a, F_a) \cdot \sum_{Y_b \in \{-1,1\}} P(F_b, Y_b|Y_a) \tag{14}$$

$$= P(Y_a)P(F_a|Y_a) \cdot \sum_{Y_b \in \{-1,1\}} P(F_b|Y_b)P(Y_b|Y_a) \tag{15}$$

$$\propto e^{Y_a \cdot F_a \beta_a} \cdot \left( m_C e^{Y_a \cdot F_b \beta_b} + (1 - m_C)e^{-Y_a \cdot F_b \beta_b} \right) \tag{16}$$

$$= m_C e^{Y_a(F_a \mu_a + F_b \mu_b)} + (1 - m_C)e^{Y_a(F_a \mu_a - F_b \mu_b)} \tag{17}$$

By putting it back to equation(11), we could get:

$$P(Y_a|F_a, F_b) = \cfrac{1}{1 + \cfrac{m_C e^{Y_a(F_a\beta_a + F_b\beta_b)} + (1-m)e^{Y_a(F_a\beta_a - F_b\beta_b)}}{m_C e^{-Y_a(F_a\beta_a + F_b\beta_b)} + (1-m)e^{-Y_a(F_a\beta_a - F_b\beta_b)}}} \tag{18}$$

The formula shows that the optimal bayes classifier depends on the non-causal factor $F_b$ given $m_C \neq 0.5$.

To give two extreme, when $m_C = 1$:

$$P(Y_a|F_a, F_b) = \frac{1}{1 + e^{2Y_a(F_a\beta_a + F_b\beta_b)}} \tag{19}$$

In this way, the optimal classifier is $\beta = [2\beta_a, 2\beta_b]^T$ for the two factors $F_a$ and $F_b$.

When $m_C = 0.5$:

$$P(Y_a|F_a, F_b) = \frac{1}{1 + e^{2Y_a(F_a\beta_a)}} \tag{20}$$

In this way, the optimal classifier is $\beta = [2\beta_a, 0]^T$, which only utilizes the first factor $F_a$ and assign zero weights for the non-causal factor $F_b$.

## A.3 Classifier trained on limited dataset

In the following we're considering the cases whether there's no task correlation in training set ($m_C = 0.5$). Though we have shown previously the optimal classifier should be invariant to non-causal factors given unlimited data, in reality with limited training dataset, the model could still utilize non-causal factors as noise.

Assume the training data contains spurious feature $S$ appended to causal feature $C$ for ground-truth linear model $Y = \theta^* C$, both under-parametrized and over-paramatrized linear model $\hat{Y} = \hat{\theta} C + \hat{\beta} S$ will assign non-zero weights $\hat{\beta}$ for spurious feature $S$.

Let $x \in \mathbb{R}^{(d+1)\times 1}$ denote the feature, where $x[1 : d] = c$ is the causal feature, and $x[d + 1] = s$ is spurious feature.

Let ground-truth linear model $y_i = f_{\theta^*}(x) = \theta^* \cdot x_i + \epsilon_i = c \cdot \theta_c^* + \epsilon_i$, where $\theta^* = [\theta_c^*, 0] \in \mathbb{R}^{(d+1)\times 1}$ and $\epsilon \sim N(0, \sigma^2)$.

Given training dataset $X \in \mathbb{R}^{n\times(d+1)}$ and $Y = X\theta^* + \varepsilon = C\theta_c^* + \varepsilon \in \mathbb{R}^{n\times 1}$, the closed-form solution $\hat{\theta} \in \mathbb{R}^{(d+1)\times 1}$ for linear regression model is:

$$\hat{\theta} = X^+ Y^+ = X^+(X\theta^* + \varepsilon) \tag{21}$$

The generalization error is:

$$\mathcal{L} = \mathbb{E}_x\left[\left((\hat{\theta} - \theta^*) \cdot x\right)^2\right] \tag{22}$$

$$= \mathbb{E}_x\left[\left((X^+X - I)\theta^* \cdot x + X^+\varepsilon \cdot x\right)^2\right] \tag{23}$$

$$= \mathbb{E}_x\left[\left((X^+X - I)\theta^* \cdot x\right)^2\right] + \sigma^2\mathbb{E}_x\left\|(X^T)^+ x\right\|_2^2 \tag{24}$$

The first term is bias and the second is variance.

If $X = [C, 0]$, which only contains causal feature without any spurious feature, we denote the learned parameter and loss as $\hat{\theta}_C$ and $\mathcal{L}_C$.

If $X = [C, S]$, which contains the spurious feature, we denote the learned parameter and loss as $\hat{\theta}_S$ and $\mathcal{L}_S$.

Our goal is to prove the learned parameter weight for the spurious feature is not zero. We'll study it in both underparamtrizied ($d + 1 \leq n$) setting, where the solution is equivalent to least-square solution; and overparametrized ($d > n$), where the solution is equivalent to min-norm solution.

### A.3.1 Underparametrized Setting

**Loss** Since $X \in \mathbb{R}^{n \times (d+1)}$ has independent column due to under parametrization assumption, we can find pseudo-inverse such that $X^+ X = I$. Thus the bias term in $\mathcal{L}$ is 0, and we only need to consider the variance term.

$$\mathcal{L}_S - \mathcal{L}_C = \sigma^2 \left( \mathbb{E}_x \left\| \begin{bmatrix} C^T \\ S^T \end{bmatrix}^+ x \right\|_2^2 - \mathbb{E}_x \left\| \begin{bmatrix} C^T \\ 0 \end{bmatrix}^+ x \right\|_2^2 \right) \tag{25}$$

Since $\|A^+ x\|_2^2 = \min_{Z:Az=x} \|z\|_2^2$, and obviously $\left\{ z \middle| \begin{bmatrix} C^T \\ S^T \end{bmatrix} z = x \right\} \subseteq \left\{ z \middle| \begin{bmatrix} C^T \\ 0 \end{bmatrix} z = x \right\}$ as the first one has one more constraint. Therefore, $\left\| \begin{bmatrix} C^T \\ S^T \end{bmatrix}^+ x \right\| \geq \left\| \begin{bmatrix} C^T \\ 0 \end{bmatrix}^+ x \right\|_2^2$, and thus $\mathcal{L}_S \geq \mathcal{L}_C$.

**weight** By the theorem 1 of (Baksalary & Baksalary, 2007), if $d+1 \leq n$, $X = [S, T] \in \mathbb{R}^{n \times (d+1)}$ has independent column, thus we have

$$X^+ = \begin{bmatrix} C^T \\ S^T \end{bmatrix}^+ = \begin{bmatrix} (I-Q)C(C^T(I-Q)C)^{-1} \\ \frac{(I-P)S}{S^T(I-P)S} \end{bmatrix} \tag{26}$$

where $P = CC^T, Q = SS^T$.

Therefore,

$$\hat{\theta}_S[d+1] = \frac{(I-P)S}{S^T(I-P)S} Y = \frac{(I-P)S(C\theta_C^* + \varepsilon)}{S^T(I-P)S} \tag{27}$$

### A.3.2 Overparametrized Setting

In this setting the closed-form solution is equivalent to minimum-norm solution, such that:

$$\hat{\theta} = \arg\min_{\theta} \|\theta\|_2^2 \tag{28}$$

$$s.t. \ X\theta = Y \tag{29}$$

**weight** Since $X$ is have full row rank, $(XX^T)^{-1}$ exists, thus we have:

$$X^+ = X^T(XX^T)^{-1} \tag{30}$$

Based on the Sherman-Morrison formula, we have:

$$(XX^T)^{-1} = (CC^T + SS^T)^{-1} = G - \frac{GSS^TG}{1 + S^TGS} \tag{31}$$

where $G = (CC^T)^{-1}, u = \frac{b^TG}{1+b^TGb}$. Therefore:

$$X^+ = \begin{bmatrix} C^T \\ S^T \end{bmatrix}^+ = \begin{bmatrix} (I-bu)C^+ \\ u \end{bmatrix} \tag{32}$$

Thus

$$\hat{\theta}_S[d+1] = \frac{b^TG}{1+b^TGb} Y = \frac{b^TG(C\theta_C^* + \varepsilon)}{1+b^TGb} \tag{33}$$

To sum up, given limited training dataset, even without spurious correlation between tasks, and non-causal features only serve as noise, the model could still learn to assign non-zero weights to non-causal features to overfit the dataset. Therefore, in MTL setting, when the number of tasks increase, the shared representation encodes many causal features from different tasks. Even without spurious correlation, it will lead to overfitting issue. And such problem could be exacerbated by spurious correlation that we show in section A.2.

## B    Synthetic Analysis of Multi-SEM with more tasks and saliency map

In section 2.2 we compare model trained by MTL with STL with two tasks. Here we show the results conducted in Multi-SEM with more than two tasks in Table 5. The results show decreasing $\text{Acc}_{val}$ and higher usage of spurious feature $\rho_{spur}$ compared with STL, with increasing number of tasks. This matches our hypothesis that MTL could incorporate more non-causal features / factors into shared representation, increasing the risk of utilizing overfitting. We also show the saliency map for each feature dimension in Figure 7. It shows that the model trained by MTL exploits non-causal features (dimension 20-120) more than the model trained by STL. All these results empirically support our claim that with spurious task correlation, model trained by MTL utilize non-causal factors more and generalize worse than STL.

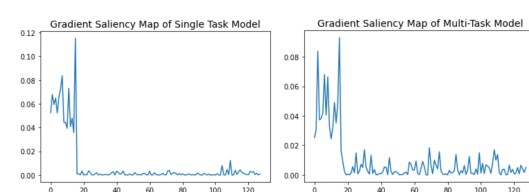

Figure 7: The gradient saliency map of Multi-SEM. The model trained by MTL exploits non-causal features (spurious) more.

| #Tasks | | 2 | 3 | 4 | 5 | 6 | 7 | 8 |
|---|---|---|---|---|---|---|---|---|
| MTL | $\text{Acc}_{val}$ | 0.846 | 0.838 | 0.824 | 0.809 | 0.785 | 0.752 | 0.719 |
| | $\rho_{spur}$ | 0.328 | 0.357 | 0.391 | 0.429 | 0.475 | 0.530 | 0.594 |
| STL | $\text{Acc}_{val}$ | 0.874 | 0.861 | 0.848 | 0.836 | 0.827 | 0.810 | 0.797 |
| | $\rho_{spur}$ | 0.261 | 0.289 | 0.314 | 0.354 | 0.385 | 0.407 | 0.435 |

Table 5: Results on Multi-SEM with more than 2 tasks.

## C    Pseudo-Code and more discussion of MT-CRL

The full psudo-code of proposed MTL is shown in Alg. 1. We first use disentangled MMoE model to calculate loss for each task $R_t(\Phi, A_t, f_t)$, and also calculate disentangled and graph regularization. We then calculate invariant regularization over train/valid split. The most important part is line 11 we detach the per-task predictors from computational graph, so that when we calculate gradient (via $loss.backward$), we only calculate gradient over graph $A$ and encoder $\Phi$.

Ideally the invariant loss should be calculated based on different environmental split, similar to what is utilized in existing Out-Of-Distribution Generalization works. However, in MTL setting, there's no datasets designed specifically for studying OOD generalization or spurious correlation. To make current approach suitable for real-world applications, we only utilize two environment split (i.e. train and valid from existing datasets).

Noted that in our framework we adopt a simple linear correlation regularization to enforce disentanglement. This regularization only forces representation to be linearly de-correlated, and a more strict solution might be reducing the mutual information (MI). However, existing methods to minimizing MI requires either knowing the latent distribution (e.g. InfoGAN. We report BetaVAE in Table 3 with similar intuition but performs worse) or over estimated MI (e.g. MINE). We indeed tried adding discriminator for every module pair and adopted Minmax training to minimize estimated MINE. The result is unstable and no better. Module output's norm is very large and only the centers are seperated rather than disentangled. Therefore, we only utilize the linear de-correlation methods that perform well in our experiments. We show the mutual correlation of every pairs of modules in Figure 8

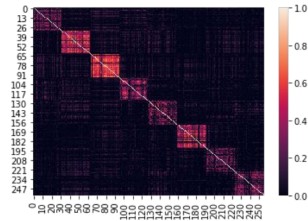

Figure 8: The heatmap of mutual correlation $\rho(\mathbf{Z}_i, \mathbf{Z}_j)$ between every pairs of modules.

learned in MultiMNIST dataset. It shows that after learning, the modules indeed learn to be linearly de-correlated between each other, and only have correlated neurons within each module.

**Algorithm 1:** Pseudo-Code of proposed MT-CRL (use $\mathcal{L}_{G\text{-}IRM}^{Var}$ as invariant regularizer)

**Require:** shared encoders with $K$ different neural modules $\Phi = \big[\Phi_i(\cdot)\big]_{i=1}^{K}$, biadjacency matrix $A = \text{sigmoid}(\theta) \in [0,1]^{T\times K}$, per-task predictors $\mathcal{F} = \{f_t\}_{t\in\mathcal{T}}$, minibatch with environment label and loss function for each task $\mathcal{B}_{t\in\mathcal{T}} = \{X_t, Y_t, E_t, \mathcal{L}_t\}_{t\in\mathcal{T}}$

1: $\mathcal{L}_{\mathcal{B}} = 0$
2: **for** each task $t \in \mathcal{T}$ **do**
3:     Get $X_t, Y_t, \mathcal{L}_t$ from $\mathcal{B}_t$
4:     $\mathbf{Z} = \big[\mathbf{Z}_i\big]_{i=1}^{K} = \big[\Phi_i(X_t)\big]_{i=1}^{K}$
5:     $\hat{Y}_t(\mathbf{X_t}) = f_t\big(\sum_i A_{t,i}\cdot \mathbf{Z}_i)\big) = f_t\big(\sum_i A_{t,i}\cdot \Phi_i(\mathbf{X_t})\big)$
6:     $\mathcal{L}_{\mathcal{B}} = \mathcal{L}_{\mathcal{B}} + R_t\big(\Phi, A_t, f_t\big) = \mathcal{L}_{\mathcal{B}} + \mathcal{L}_t(\hat{Y}_t(\mathbf{X_t}), Y_t)$
7:     $\mathcal{L}_{\mathcal{B}} = \mathcal{L}_{\mathcal{B}} + \mathcal{L}_{decor}(\Phi)_t = \mathcal{L}_{\mathcal{B}} + \lambda_{decor}\cdot \sum_{i=1}^{k}\sum_{j=i+1}^{k}\big\|\rho\big(\Phi_i(X_t),\Phi_j(X_t)\big)\big\|_F^2$
8: **end for**
9: $\mathcal{L}_{\mathcal{B}} = \mathcal{L}_{\mathcal{B}} + \mathcal{L}_{graph}(A) = \mathcal{L}_{\mathcal{B}} + \Big(\lambda_{sps}\cdot \|A\|_1 - \lambda_{bal}\cdot \text{Entropy}\big(\frac{\sum_t A_{t,*}}{\sum_{t,i} A_{t,i}}\big)\Big)$
10: $grad = \nabla_{A,\mathcal{F},\Phi}\,\mathcal{L}_{\mathcal{B}}$
11: Detach $\mathcal{F} = \{f\}_t$ from computational graph (use $tf.stop\_gradient$ or $torch.zero\_grad$)
12: $\mathcal{L}_{G\text{-}IRM}^{Var}(\Phi, A|f) = 0$
13: **for** each task $t \in \mathcal{T}$ **do**
14:     Get environment label $E_t$. In our experimental setting it's train and valid set.
15:     $\mathcal{L}_{G\text{-}IRM}^{Var} = \mathcal{L}_{G\text{-}IRM}^{Var} + \sum_{e\in E_t}\frac{1}{|E_t|}\big\|\nabla_{A=A_t}R_t^e\big(\Phi, A, f_t\big) - \text{Avg}_e\big(\nabla_{A=A_t}R_t^e\big)\big\|^2$
16: **end for**
17: $grad = grad + \nabla_{A,\Phi}\,\mathcal{L}_{G\text{-}IRM}^{Var}(\Phi, A|f)$
18: Use optimizer to update the model via gradient $grad$

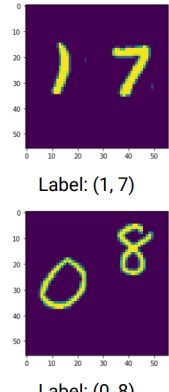

Label: (1, 7)

Label: (0, 8)

- We put one digit on left, another digit on right
- In training set, the label of two digit have correlation.
- In test set, the correlation is reversed.

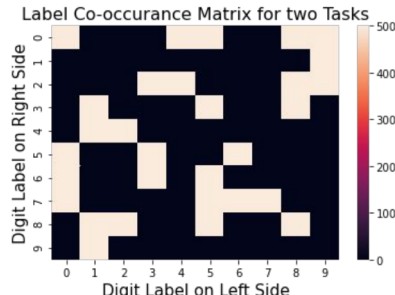

Label Co-occurance Matrix for two Tasks

Figure 9: Illustrative figure for spurious Multi-MNIST dataset used for analaysis.

## D Details about Dataset

### D.1 Synthetic Datasets

**Multi-SEM.** We mostly follow the setting of linear Structural Equation Model (SEM) proposed by Rosenfeld et al. (2021). The two binary-classification task labels $Y_a$ and $Y_b$ are causally related to two distinctive factors $F_a$ and $F_b$ respectively via Gaussian distribution. We define the spurious correlation of the two labels by the probability that the two labels are the same: $C_{dist}^{MTL} = P(Y_a = Y_b)$. We set different $C_{dist}$ for training and test sets to simulate distribution shifts.

**Multi-MNIST.** We modified the multi-digit MNIST (Sun, 2019), which samples two digit pictures and put in left and right position. The generative variables $F_{left}, F_{right}$ are the digit images and data input is simply their concatenation: $X = [F_{left}, F_{right}]$. We define the task correlation $C_{dist}^{MTL}$ by co-occurrence probability of the two digit labels. We randomly shuffle the label pairs and split the

training and test set such that the class label pairs do not overlap. An illustrative data point and the label pairs in training set is shown in Figure 9.

### D.2   Real-world Datasets

**Multi-MNIST** (Harper & Konstan, 2016) is a multi-task variant of MNIST dataset, which samples two digit pictures and put in left and right position. We mainly modified from the this code repo[2] to generate the dataset. We sample 10,000 images for each label pair, so totally there are 1M data samples. As discussed in analysis section, to mimic distribution shifts (i.e., task correlation $C_{dist}^{MTL}$), we randomly shuffle the label pairs and split the train, valid and test set with ratio 3:1:1, such that every image co-occurrence correlation will no longer appear again in test set. We utilize the same CNN architectures and hyperparameter adopted in Yu et al. (2020) as base encoder, and one-layer MLP as per-task predictor.

**MovieLens** (Harper & Konstan, 2016) is a Movie recommendation dataset that contains 10M rating records[3] of 10,681 movies by 71,567 users from Jan. 1996 to Dec. 2008. We consider the rating regression for movies in each genre as different tasks. There are totally 18 different genres, including Action, Adventure, Animation, Children's, Comedy, Crime, Documentary, Drama, Fantasy, Film-Noir, Horror, Musical, Mystery, Romance, Sci-Fi, Thriller, War and Western. To mimic distribution shifts across train, valid and test set, we split the data based on timestamp with ratio 8:1:1, and filter out non-overlapping users and movies from each set. We utilize a embedding layer followed by two-layer MLP as base encoder, and one-layer MLP as per-task predictor.

**Taskonomy** (Zamir et al., 2018) is a large-scale MTL benchmark dataset of indoor scene images from various buildings[4]. Every image has annotations for a set of diverse computer vision tasks. We follow the setting of (Balaji et al., 2020) to use 8 tasks, including curvature estimation, object classification, scene classification, surface normal estimation, semantic segmentation, depth estimation, occlusion edge, 2D keypoint estimation and 3D keypoint estimation. For these tasks, object and scene classification tasks are trained using cross entropy loss, semantic segmentation using pixelwise cross entropy, curvature estimation using L1 loss, and all other tasks using L2 loss. To mimic distribution shift, we select images from non-overlapping 48, 3, 3 buildings as train, valid and test set. The total training size is 324864 samples. We use Resnet-50 model as our base encoder network, and 15-layer CNN model with upsampling blocks as the per-task predictor.

**NYUv2** (Silberman et al., 2012) is a dataset of 1449 RGB-D indoor scene images[5] with three tasks: 13-class semantic segmentation, depth estimation, and surface normals prediction. We use mean Intersection-Over-Union (mIoU), Relative Error (Rel Err) and Angle Distance as evaluation metric for the three tasks respectively. To mimic distribution shift, we split the dataset by scene labels into train, valid and test set with ratio 8:1:1. We follow the setting adopted in Yu et al. (2020) to use Segnet (Badrinarayanan et al., 2015) as the base encoder.

**CityScape** (Cordts et al., 2016) is a dataset of street-view images[6] with two tasks: semantic segmentation and depth estimation. We use mean Intersection-Over-Union (mIoU) and Relative Error (Rel Err) as evaluation metric for the three tasks respectively. We follow the same data pre-processing procedure of the original paper, and split images based on city into 2475, 500 and 500 train, valid and test samples. We follow the setting adopted in Yu et al. (2020) to use Segnet (Badrinarayanan et al., 2015) as the base encoder.

| Tasks (Metric) | STL | MTL | PCGrad | GradVac | DANN | IRM | MT-CRL + $\mathcal{L}_{G\text{-}IRM}^{Var}$ |
|---|---|---|---|---|---|---|---|
| Left-Digit (Acc.) | $0.871 \pm 0.018$ | $0.844 \pm 0.019$ | $0.880 \pm 0.019$ | $0.884 \pm 0.017$ | $0.878 \pm 0.020$ | $0.887 \pm 0.010$ | $0.912 \pm 0.018$ |
| Right-Digit (Acc.) | $0.877 \pm 0.015$ | $0.848 \pm 0.018$ | $0.888 \pm 0.017$ | $0.886 \pm 0.018$ | $0.884 \pm 0.017$ | $0.889 \pm 0.015$ | $0.918 \pm 0.019$ |

Table 6: Results for Multi-MNIST dataset.

The results show that a middle number of module ($K$=8) achieves the best performance under the same size of model. Note that disentangled representation learning methods like BetaVAE assume

---

[2] https://github.com/shaohua0116/MultiDigitMNIST
[3] https://files.grouplens.org/datasets/movielens/ml-10m.zip
[4] http://taskonomy.stanford.edu/
[5] https://cs.nyu.edu/~silberman/datasets/nyu_depth_v2.html
[6] https://www.cityscapes-dataset.com/

| Metric | STL | MTL | PCGrad | GradVac | DANN | IRM | MT-CRL + $\mathcal{L}_{G\text{-}IRM}^{Var}$ |
|---|---|---|---|---|---|---|---|
| Avg. MSE | $0.894 \pm 0.006$ | $0.892 \pm 0.005$ | $0.892 \pm 0.006$ | $0.891 \pm 0.005$ | $0.890 \pm 0.007$ | $0.890 \pm 0.004$ | $0.884 \pm 0.006$ |

Table 7: Results for MovieLens dataset.

| Tasks (Metric) | STL | MTL | PCGrad | GradVac | DANN | IRM | MT-CRL + $\mathcal{L}_{G\text{-}IRM}^{Var}$ |
|---|---|---|---|---|---|---|---|
| object classification (Cross Entropy) | 3.37 | 3.18 | 3.09 | 3.06 | 3.13 | 3.16 | 3.01 |
| scene classification (Cross Entropy) | 2.65 | 2.59 | 2.54 | 2.51 | 2.58 | 2.59 | 2.47 |
| semantic segmentation (Cross Entropy) | 1.68 | 1.54 | 1.47 | 1.49 | 1.53 | 1.56 | 1.43 |
| curvature estimation (L1 Loss) | 0.246 | 0.224 | 0.218 | 0.212 | 0.237 | 0.226 | 0.208 |
| surface normal estimation (L2 Loss) | 0.138 | 0.141 | 0.136 | 0.139 | 0.152 | 0.150 | 0.125 |
| occlusion edge detection (L2 Loss) | 0.134 | 0.138 | 0.132 | 0.133 | 0.137 | 0.141 | 0.128 |
| 2D keypoint estimation (L2 Loss) | 0.176 | 0.171 | 0.167 | 0.163 | 0.169 | 0.168 | 0.158 |
| 3D keypoint estimation (L2 Loss) | 0.194 | 0.205 | 0.199 | 0.196 | 0.204 | 0.201 | 0.191 |

Table 8: Results for Taskonomy dataset.

| Tasks (Metric) | STL | MTL | PCGrad | GradVac | DANN | IRM | MT-CRL + $\mathcal{L}_{G\text{-}IRM}^{Var}$ |
|---|---|---|---|---|---|---|---|
| Segmentation (mIoU) | 13.27 | 17.64 | 19.64 | 19.68 | 17.12 | 17.54 | 19.81 |
| Depth (Rel Err) | 0.653 | 0.651 | 0.591 | 0.593 | 0.637 | 0.639 | 0.585 |
| Surface Normal (Angle Distance) | 35.18 | 31.52 | 30.98 | 31.04 | 31.69 | 32.03 | 30.85 |

Table 9: Results for NYU-V2 dataset.

| Tasks (Metric) | STL | MTL | PCGrad | GradVac | DANN | IRM | MT-CRL + $\mathcal{L}_{G\text{-}IRM}^{Var}$ |
|---|---|---|---|---|---|---|---|
| Segmentation (mIoU) | 50.87 | 51.63 | 52.84 | 52.76 | 51.91 | 52.05 | 53.12 |
| Depth (Rel Err) | 33.85 | 32.75 | 32.12 | 32.08 | 32.71 | 32.64 | 31.86 |

Table 10: Results for CityScape dataset.

| $K$ | 1 | 2 | 4 | **8** | 16 | 32 | 64 | 128 |
|---|---|---|---|---|---|---|---|---|
| Acc. | 0.824 | 0.897 | 0.904 | **0.915** | 0.911 | 0.902 | 0.893 | 0.882 |

Table 11: Hyperparameter tuning results for number of module ($K$) over Multi-MNIST dataset.

that every dimension is mutually independent ($K = 128$ in our case), which restricts the model capacity. Therefore, a middle K is a trade-off between model disentanglement and capacity. In all other datasets, we just use $K = 8$ by default and didn't do further tuning.

# E  Detailed Results on each Dataset

We report the performance on each task for the five benchmark datasets in Table 6-10. We use 8 GPU to run each experiments. As shown in the tables, the scale of different task's evaluation metric differ a lot, and thus in the main paper we adopt relateive performance improvement compared to vanilla MTL to evaluate each method. Nevertheless, our MT-CRL with invariance regularization could achieve the best results over nearly all the tasks.

# F  Determine the number of modules $K$

$K$ is a hyperparameter that could be tuned. To control the same model complexity, we fix the output dimension $d$, i.e., 128, and then each module's dimension is $\frac{d}{K}$. We show the results on Multi-MNIST with different $K$ in Table 11.

The results show that a middle number of module (K=8) achieves the best performance under the same size of model. In all other datasets, we just use $K = 8$ by default and didn't do further tuning to test our method's generalization capacity, this could avoid the situation that the final performance improvement is mainly caused by extensive hyper-parameter tuning.

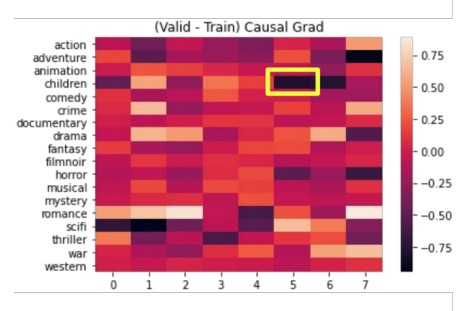

View 1: Shine, go, shawshank, psycho, dumber
View 2: Rocky, october, casino, muppet, payback
View 3: forrest, gump, carrie, now, saving
View 4: i, house, monty, at, life, dark
**View 5: good, club, young, stripes, die**
View 6: 1978, out, witness, shining, chocolate
View 7: space, la, love, best, graduate
View 8: die, life, black, true, amistad

Figure 11: (valid-train) Task-to-Module gradients of model **without** MT-CRL on MovieLens.

**Capacity-Disentanglement Tradeoff**   Noted that with a fixed number of dimension $d$, with larger $K$, the model capacity is reduced. The widely adopted disentangled representation learning methods like BetaVAE mostly assume that every dimension is mutually independent ($K = d$ in our case), which restricts the model capacity to extreme case. And the results in Table 3 also show that our current disentangled approach performs empirically better than BetaVAE. One potential reason is that we choose allow a middle K is a trade-off between model disentanglement and capacity, in which only the dimension across the block is de-correlated, why the ones within block could still correlated, so as to maintain model expressiveness.

Note that the optimal choice of $K$ should ideally should be proportional to number of true generative factors that are related to downstream tasks. Therefore, for dataset with a large amount of tasks, we should choose larger number of $K$, and also consider increasing the total number of dimension $d$ to increase the model capacity while maintaining disentanglement. In our paper for large dataset such as Taskonnomy we didn't do further tuning due to limited resources, so the performance could be potentially further improved, which we leave for future exploration.

## G   More Case Studies to show MT-CRL can alleviate spurious correlation

Here we show more details about the case study we conduct for analyzing how MT-CRL could alleviate spurious correlation, as a complementary.

As introduced in case study, we use the task-to-module gradients $\frac{\partial (f(\Phi(x))[y])}{\partial F}$ to illustrate how each task utilize each module. We could utilize the (valid-train) score to show which module is used by training set but not helpful for valid set, meaning it is spurious.

We first show detailed results on MovieLens dataset. As shown in Figure 11, without MT-CRL, there eixst many modules assigned negative (valid-train) causal grad, as shown in the color bar. Among them, view 5 is mostly inconsistent with the training results as we show in case study. Also, the other modules' key words are also not very accurate to describe the properties of each movie.

After we add MT-CRL, as shown in Figure 12, all of the modules receive positive (valid-train) causal grad, meaning that they either not utilized in training stage, or every used modules are still useful in valid stage. In addition, all the modules' key words are much more accurate to describe each type of movie than before.

In addition, our learned task biadjacency graph $A$ could also be used to describe the similarity between task. If two tasks share more causal feature, they are more similar. We thus cauculate the task-averaged score of $A$, and plot a sparse smilarity graph in Figure 10. It shows that our MT-CRL could learn to group similar types of adult movies, such as war,

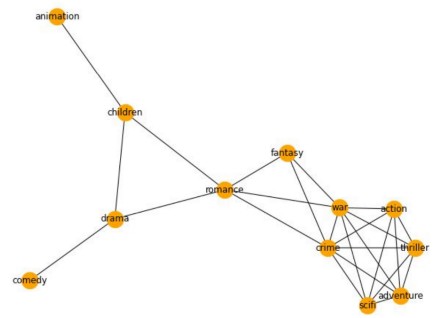

Figure 10: The task similarity induced by causal graph $A$ for MovieLens dataset (threshold = 0.1).

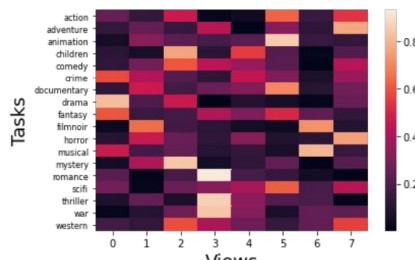
(Drama) View 0: amadeus, amistad, farewell, thunderball
(Filmnoir) View 1: spartacus, bad, miracle, croupier
(Mystery) View 2: Werewolf, serpico, wrath, hunt
(Romance) View 3: Wives, Sister, Guys, Titanic
(Children) View 4: Pink, Parenthood, Alice, Jungle
(Animation) View 5: Titans, apollo, dancing, willy
(Musical) View 6: singers, chuck, arlington, lovers
(Adventure) View 7: cube, walking, benjamin, felicia

Figure 12: (valid-train) Task-to-Module gradients of model **with** MT-CRL on MovieLens.

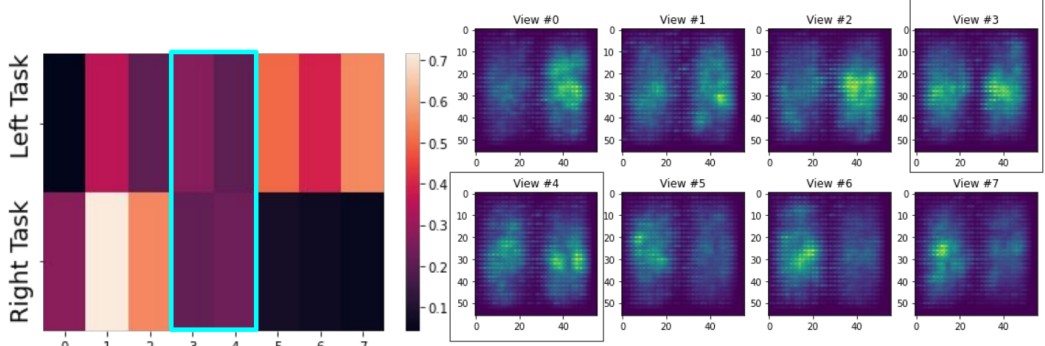

Figure 13: (valid-train) Task-to-Module gradients of model **without** MT-CRL on Multi-MNIST.

crime, thriller, adventure into the same group in the right down part, and link children-friendly movies, such as animation and children together. romance movie is a link between adult cluster and children movie. This similarity graph matches our human expectation, showing that our learned causal graph indeed help similar group use similar causal features.

We then show the (valid-train) Task-to-Module gradients over Multi-MNIST datasets. It is very apparent that the two digit classifier doesn't share any overlapping causal features. However, as shown in Figure 13, without MT-CRL, the model still learns to assign similar weights to module 3 and 4. This is also illustrated by the gradient saliency map for each module. Module 3 and 4 have high attention on both left and right side.

With MT-CRL, in Figure 14, the learned task-to-module assignment is much sparse and clear. Also each module's saliency map only focus on one side of pixels. By looking at each task output's saliency map, in Figure 15, we can see the model with MT-CRL can help to focus only on causal part, compared with Figure 3(b) that have high weights on both.

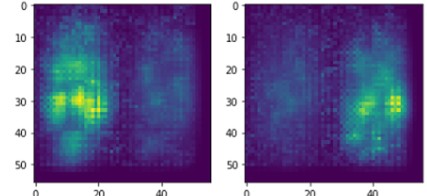

Figure 15: The gradient saliency map of left-digit and right-digit classifier trained via MT-CRL. Compared with Figure 3(b), MT-CRL indeed helps alleviate spurious correlation.

Both the MovieLens and Multi-MNIST case studies show that MT-CRL could help alleviate spurious correlaiton issue. For the other datasets, such as Taskonomy, NYUv2 and CityScape, their task output layer is very different and thus it's hard to get normalized gradient to show in one figure. In the future, we plan to do more thorough analysis by manually label several spurious feature or environmental groups, and design better methods to visualize how MTL model utilize non-causal features.

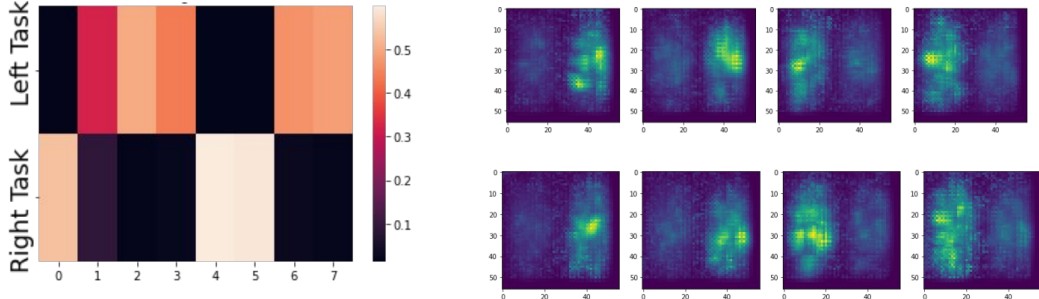

Figure 14: (valid-train) Task-to-Module gradients of model **with** MT-CRL on Multi-MNIST.

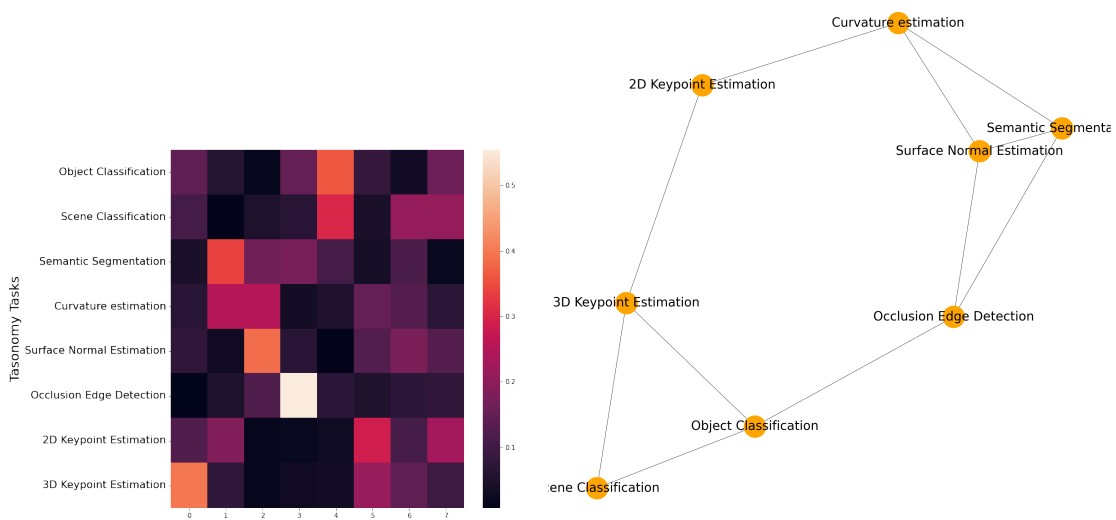

Figure 16: Task-to-Module Routing Graph ($A$) of model trained on Taskonomy dataset.

Figure 17: Task Similarity graph induced by Task-to-Module Graph $A$ for Taskonomy dataset (threshold = 0.1).

**Case Study on Taskonomy.** In addition, we show the Task-to-Module routing graph and also the induced task similarity graph of Tasknomy dataset. As is shown in the figure, some similar task like 2D keypoint Estimation and 3D keypoint are liked together, and also the hard task like semantic segmentation receives information from curvature estimation, surface normal estimation and occlusion edge detection. These findings fit the observation of original Taskononmy analysis (Zamir et al., 2018). As stated in the limitation, we leave the deeper analysis in Tasknonmy about spurious feature as future work, as currently we don't have the ground-truth anotation about which part of image input is causally related to each task.

## H Detailed Hyper-parameter Selection Procedure and Sensitivity Analysis

Here we introduce the procedure and results of hyper-parameter tuning.

Before discussing hyper-parameter selection, let me explain our baseline setup and experiment setting again. We have a single validation set, potentially bringing OOD to the training set. Our method only uses the training set to calculate the loss to update both encoder and per-task predictors. We then use the hold-out validation set to calculate the loss w.r.t graph weights A, and update it via invariance regularization. This avoids overfitting the validation set.

We mainly split the hyper-parameters into two sets:

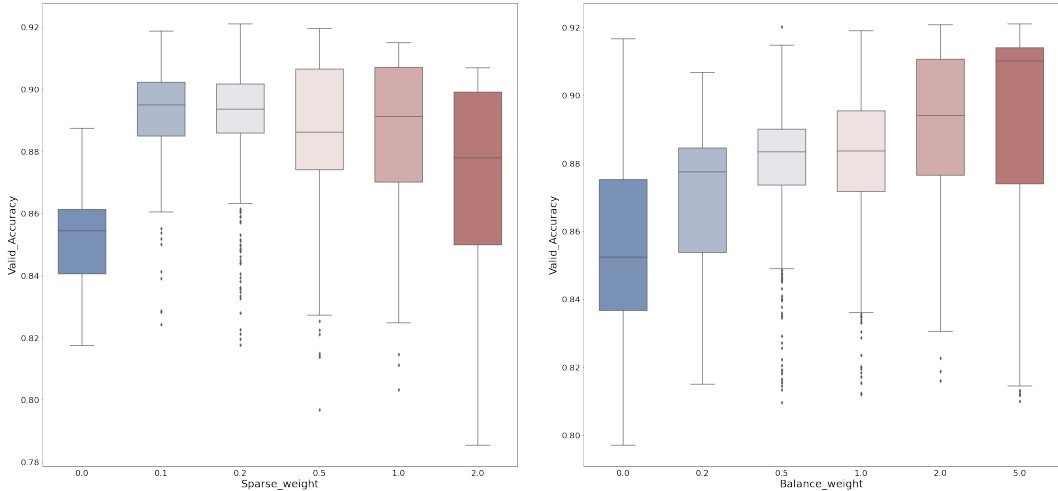

Figure 18: Hyper-parameter tuning results for sparse weight ($\lambda_{sps}$) on Multi-MNIST.

Figure 19: Hyper-parameter tuning results for balancing weight ($\lambda_{bal}$) on Multi-MNIST.

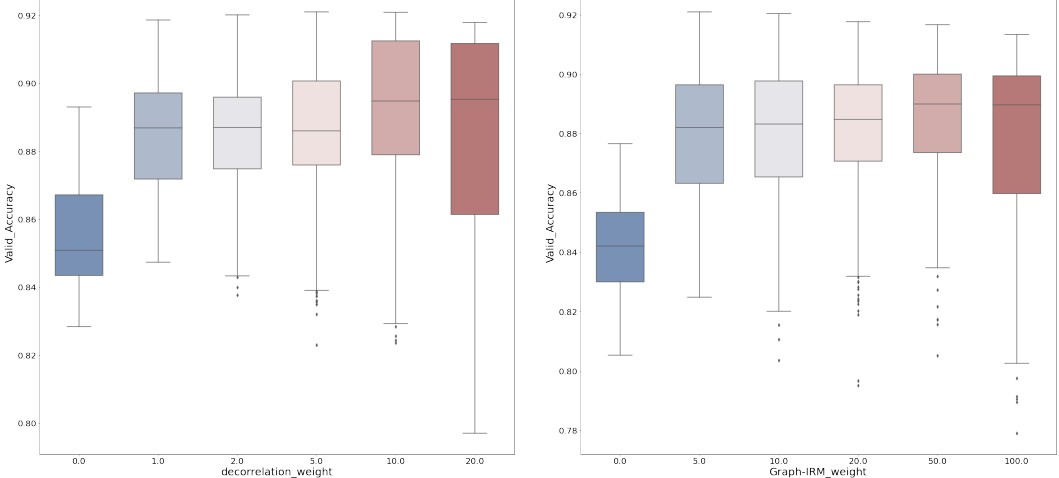

Figure 20: Hyper-parameter tuning results for disentanglement weight ($\lambda_{decor}$) on Multi-MNIST.

Figure 21: Hyper-parameter tuning results for correlation weight ($\lambda_{G-IRM^{Var}}$) on Multi-MNIST.

- General hyper-parameter related to all baselines (including ours), including number of hidden states, optimizer, learning rate, number of modules ($K$);

- Our MT-CRL specific hyper-parameter, including weights for disentanglement ($\lambda_{decor}$), sparsity ($\lambda_{sps}$), balance ($\lambda_{bal}$), and invariance ($\lambda_{G-IRM}$).

For both sets of hyper-parameters, we only tune on the same OOD validation set we used for our method. All hyper-parameters except $K$ are standard hyperparameters for the MTL model. For datasets CityScape, NYUv2, and Taskonomy, we directly use the reported hyperparameter and dataset setting in previous papers (Yu et al., 2020; Balaji et al., 2020), in order to achieve a fair comparison. For MultiMNIST and MovieLens, we conduct a grid search for basic parameters, including the number of layers, number of hidden dimensions, optimizer, and learning rate, on the Vanilla MMoE MTL model without regularization. For the number of module ($K$), please refer to Sec F. After we determine these general hyper-parameters, we fix them and use them for all different MTL methods. This makes the comparison fair and ensures our performance improvement is not due to extensive hyper-parameter tuning of our method.

| Methods | Multi-MNIST Accuracy | MovieLens MSE |
|---|---|---|
| MT-CRL with $\mathcal{L}_{G\text{-}IRM}^{Var}$ and **default** hyper-parameter | $0.915 \pm 0.018$ | $0.884 \pm 0.006$ |
| MT-CRL with $\mathcal{L}_{G\text{-}IRM}^{Var}$ and **randomly chosen** hyper-parameter | $0.904 \pm 0.021$ | $0.887 \pm 0.006$ |
| Vanilla MTL baseline | $0.846 \pm 0.018$ | $0.892 \pm 0.005$ |

Table 12: Results on Multi-MNIST and MovieLens with a randomly chosen set of hyper-parameter.

Next, we tune the MT-CRL-specific hyper-parameters on the validation set. We think this setting is reasonable as we didn't utilize this validation set to calculate training loss. Thus it could still be regarded as a whole-out set for most model parameters (except the graph weights, which only take a tiny portion of the whole model). Note that four regularization weight terms exist to be tuned, which is many burdens for the model. Therefore, we only use Multi-MNIST, the smallest dataset in all our testbeds, to conduct hyperparameter tuning for the ML-CRL-specific hyperparameters with grid-search. This is definitely not the best choice, and tuning for each dataset could potentially improve our performance further, but that only makes our improvement higher while not changing the main conclusion of this paper. Specifically, we choose the several ranges for the four regularization weights:

- Sparse weight ($\lambda_{sps}$): $[0.0, 0.1, 0.2, 0.5, 1.0, 2.0]$
- Balancing weight ($\lambda_{bal}$): $[0.0, 0.2, 0.5, 1.0, 2.0, 5.0]$
- Disentanglement weight ($\lambda_{decor}$): $[0.0, 1.0, 2.0, 5.0, 10.0, 20.0]$
- Invariance weight ($\lambda_{G-IRM^{Var}}$): $[0.0, 5.0, 10.0, 20.0, 50.0, 100.0]$

These ranges are selected by running a few samples to determine the maximum value that should be within this range, and we keep each selection list to be a length of 6. We report the boxplot of detailed results for each regularization weights in Figure (18-21). As is illustrated, for all the regularization, using it is better than not using it ($\lambda = 0$), showing their advantage in making our MT-CRL pipeline works. We then select the optimal hyperparameter that achieves the highest validation accuracy, which is $\lambda_{sps} = 0.2, \lambda_{bal} = 5.0, \lambda_{decor} = 20, \lambda_{G-IRM} = 5.0$. Again, this selection might not be the optimal solution; for example, the tendency for $\lambda_{bal}$ seems to increase with higher, and $\lambda_{G-IRM}$ might have a better choice within the range $[0 - 5]$. However, we did not conduct more searching and used this setup. After getting such a combination of MT-CRL-specific hyper-parameter, we fix it and use it for all other larger datasets, which assume our framework with this hyper-parameter combination is consistently effective. Further tuning them on a dedicated dataset should potentially bring better performance, but we did not do it to avoid the performance improvement brought by extensive tuning.

**Sensitivity Analysis.** From the curve and also the definition of these regularization, we know that for all other terms except sparsity regularization $\lambda_{sps}$, increasing the regularization weight and strictly force model to be balance, de-correlated or invariant doesn't harm too much to the model training (trend didn't go down even with relatively large weight). The only exception is the sparsity regularization. With high $\lambda_{sps}$ implemented as $L_1$ loss over adjacency weights will force all to be zero, which is very harmful to model training, which is why by default, we choose the value as $\lambda_{sps} = 0.2$.

To give a simple example of whether our model is sensitive to an inappropriate setting of hyper-parameter, we run experiment on MultiMNIST and MovieLens, with the following randomly chosen hyper-parameter setting: $\lambda_{sps} = 2.0, \lambda_{bal} = 1.0, \lambda_{decor} = 2.0, \lambda_{G-IRM^{Var}} = 100.0$. The results compared with the original results are as shown in Table 12.

Note that with a randomly chosen hyper-parameter, the results on the two datasets drop slightly but are still significantly higher than the Vanilla MTL baseline. This is an informal showcase of our method's generality and not very sensitive to hyper-parameter selection.