# OpenReview forum: "Improving Multi-Task Generalization via Regularizing Spurious Correlation"
_NeurIPS.cc/2022/Conference — NeurIPS 2022 Accept_

### Official Review · Reviewer_nWqE · 2022-07-07

**Rating:** 7
**Confidence:** 3
**Soundness:** 3 good
**Presentation:** 3 good
**Contribution:** 3 good

**Summary:**

This work propose Multi-Task Causal Representation Learning framework to alleviate potential spurious correlation in Multi-task learning. Authors use three regularization methods to obtain the correct disentangled factors for experts, the sparsity for the Task-to-Module graph and the invariant causal representation respectively, which based on the Multi-gate Mixture-of-Experts architecture. Experiments on various Multi-task benchmarks and sufficient ablation studies have empirically proved the effectiveness of the method.

**Questions:**

How to determine and select the hyperparameters is reasonable when introducing so many hyperparameters in the method?
Could authors give a more explicit discussion about the conflict between distinctive and complex task-predictors and the invariant?

**Limitations:**

Not relying on the MMOE architecture may have more impact.

**Strengths And Weaknesses:**

Originality: Authors study the problem of spurious correlation existing in multitask learning which has been noticed but rarely addressed directly to the best of my knowledge. Authors combine many methods to implement their claim. While neither their method is novel, their combination is novel and purposeful. And they clearly identify related work and place it in context in their paper.

Clarity: The paper is very well written; clear, precise, and easy to understand. I was able to follow the main contributions without issues. The paper contains many various and multi-level experiments which well prove the effectiveness of their method.

Significance: Alleviating spurious correlation is an interesting direction to multitask learning and may be the key to solving negative transfer. Experimental results show that this method has a general improvement over all datasets, but not significant improvement in natural CV datasets.

Authors discuss the hyperparameters tuning in the appendix. Among all the hyperparameters, I care about the number of experts $K$ that determines how many factors the feature extractor learned, on the one hand, determines the difficulty of training. Authors just use $K = 8$ by default and don’t tune it. I think it is beneficial and important to further discuss whether imposing disentangle regularization with fixed $K$ will reduce the capacity of the model, and whether $K$ should be adjusted according to the complexity of the dataset. Adding the empirical risks of methods on training datasets maybe helpful for comparing the ability of the fitting ability. And some hyperparameters $\lambda_{decor},\lambda_{sps},\lambda_{bal}$ are not discussed, did I miss anything?

And I have an issue on $\mathcal{L}^{Var}_{G-IRM}$. The intuition behind this penalty is in conflict with the fact that distinctive and complex task-predictors are necessary for multitask learning. I'm not an expert in this area so please let me know if I've misunderstood something.

---

> ### Author Response · Authors · 2022-08-02
> **Response to Reviewer nWqE [1/2]**
>
> Thanks to the reviewer for recognizing the spurious correlation in MTL problem we studied is important and might be the key to solve negative transfer. The reviewer asks to clarify our experimental setting and model design, which we answer as follow:
>
> **1. How to determine and select the hyper-parameters is reasonable when introducing so many hyper-parameters in the method?**
>
> **Answer**: We added the detailed procedure of hyper-parameter selection and results in **Appendix H**. Here we highlight the procedure and results of hyper-parameter tuning.
>
> Before discussing hyper-parameter selection, let me explain our baseline setup and experiment setting again. We have a single validation set, potentially OOD to the training set. Our method only uses the training set to calculate the loss to update both encoder and per-task predictors. We then use the hold-out validation set to calculate the loss w.r.t graph weights A, and use it to calculate invariance regularization (G-IRM). This avoids overfitting the validation set.
>
>
> We mainly split the hyper-parameters into two sets:
> 1) General hyper-parameter related to all baselines (including ours), including number of hidden states, optimizer, learning rate, number of modules ($K$)
> 2) Our MT-CRL specific hyper-parameter, including weights for disentanglement ($\lambda_{decor}$), sparsity ($\lambda_{sps}$), balance ($\lambda_{bal}$), and invariance ($\lambda_{G-IRM}$)
>
>
> For both sets of hyper-parameters, we tune on the same OOD validation set we used for our method.
> All hyper-parameters except $K$ are standard hyper-parameters for the MTL model. For datasets CityScape, NYUv2, and Taskonomy, we directly use the reported hyper-parameter and dataset setting in previous papers [1,2], in order to achieve a fair comparison. For MultiMNIST and MovieLens, we conduct a grid search for basic parameters, including the number of layers, number of hidden dimensions, optimizer, and learning rate, on the Vanilla MMoE MTL model without regularization. For the number of module ($K$), we only tune on MultiMNIST dataset, and find the optimal value $K=8$ and use for all other datasets (please refer to **Appendix F** for more details). After we determine these general hyper-parameters, we fix them and use them for all different MTL methods. This makes the comparison fair and ensures our performance improvement is not due to extensive hyper-parameter tuning of our method.
>
>
>
> Next, we tune the MT-CRL-specific hyper-parameters on the validation set. Again, we utilize the same OOV validation set to conduct hyper-parameter tuning, as this validation set is not used to calculate training loss and be used to update encoder and per-task predictors. Only the Task-to-Module graph weights receives invariance signal, which only take a tiny portion of the whole model. (If we observe datasets from multiple environments, it would be a better choice to hold out a few environments to serve as the validation set.)
>
>
> Note that four regularization weight terms exist to be tuned, which has a large search space, especially for large datasets. Therefore, we only use Multi-MNIST, the smallest dataset in all our testbeds, to conduct hyper-parameter tuning for the ML-CRL-specific hyper-parameters with grid-search. This is definitely not the best choice, and tuning for each dataset could potentially improve our performance further, but that only makes our improvement higher while not changing the main conclusion of this paper. Specifically, we choose the several ranges for the four regularization weights:
>
> -  Sparse weight  ($\lambda_{sps}$): [$0.0, 0.1, 0.2, 0.5, 1.0, 2.0$]
> -  Balancing weight  ($\lambda_{bal}$): [$0.0, 0.2, 0.5, 1.0, 2.0, 5.0$]
> -  Disentanglement weight  ($\lambda_{decor}$): [$0.0, 1.0, 2.0, 5.0, 10.0, 20.0$]
> -  Invariance weight  ($\lambda_{G-IRM^{Var}}$): [$0.0, 5.0, 10.0, 20.0, 50.0, 100.0$]
>
> These ranges are selected by running a few samples to determine that the maximum value that should be within this range, and we keep each selection list to be a length of 6.
> We report the boxplot of detailed results for each regularization weights in Figure 17-20 in the appendix. As is illustrated, for all the regularization, using it is better than not using it ($\lambda=0$), showing their advantage in making our MT-CRL pipeline works. We then select the optimal hyperparameter that achieves the highest validation accuracy, which is $\lambda_{sps}=0.2,\lambda_{bal}=5.0, \lambda_{decor}=20, \lambda_{G-IRM}=5.0$. Further tuning them on a dedicated dataset should potentially bring better performance, but we did not do it to avoid the possibility that our method's performance improvement is brought by extensive hyper-parameter tuning.
>
> [1] The effectiveness of memory replay in large-scale continual learning
>
> [2] Gradient Surgery for Multi-Task Learning

---

> > ### Author Response · Authors · 2022-08-02
> > **Response to Reviewer nWqE [2/2]**
> >
> > (continue the first question about hyper-parameter tuning)
> >
> > We also add a simple sensitivity analysis by choosing a randomly chosen hyper-parameter combination $\lambda_{sps}=2.0,\lambda_{bal}=1.0, \lambda_{decor}=2.0, \lambda_{G-IRM^{Var}}=100.0$. The results are as below:
> >
> > |   Methods   | Multi-MNIST Accuracy | MovieLens MSE |
> > |:--------:|:------:|:------:|
> > |MT-CRL with $\mathcal{L}_{G\text{-}IRM}^{Var}$ and **default** hyper-parameter|0.915 $\pm$ 0.018 |  0.884 $\pm$ 0.006|
> > |MT-CRL with $\mathcal{L}_{G\text{-}IRM}^{Var}$ and **randomly chosen** hyper-parameter|0.904 $\pm$ 0.021 |  0.887 $\pm$ 0.006|
> > |Vanilla MTL baseline|0.846 $\pm$ 0.018 |0.892 $\pm$ 0.005|
> >
> >
> > Randomly chosen setting are lower than using default setting, but still significantly higher than the vanilla MTL model. This is an informal showcase of our method's generality and not very sensitive to hyper-parameter selection. If time permits, we plan to add more combinations of these hyper-parameters and component choices on more datasets to make our ablation study more convincing.
> >
> > **2. conflict between distinctive and complex task-predictors and the invariant.**
> >
> > **Answer**: I think the reviewer might misunderstand what we did for $L_{G-IRM}^{Var}$.
> >
> > In short, our method is designed specifically for allowing distinctive and complex task predictors. We only regularize the “causal graph weights” to be invariant across different environments (between train and valid set in our experiments) for each task, not across tasks, so that it does not restrict task predictors.
> >
> > The original IRM regularization assumes a single all-one predictor to be optimal, which is not applicable for MTL, as each task has very different and complex predictors. To alleviate this issue, we propose only to optimize the graph weights. When we calculate the invariance score (in eq 8 and 9), we only calculate the loss over the causal graph weights (A), which only have parameter #module * #task and are not likely to overfit. Pseudocode is shown in Alg1 in Appendix, and I hope it can solve your question.
> >
> > In addition, another potential issue is that if the number of training samples (not number of environment) for each environment is too small, a complex predictor could easily overfit all training samples and get near-zero gradients. In our experiments, to avoid this issue, we only calculate the loss over the causal graph weights (A), which only have parameter #module * #task and are not likely to overfit. As reviewer vBSD points out, a similar trick has been adopted in eq 6 in [1].
> > In addition, another potential issue is that if the number of training samples (not the number of environments) for each environment is too small, the predictor could easily overfit all training samples and get near-zero gradients. In our experiments, we only consider a training set and a validation set as two environments to avoid this issue. We only take the training set to calculate the loss to update both encoder and per-task predictors. We then use the hold-out validation set to calculate the loss w.r.t graph weights A, and update it via invariance regularization. (This avoids overfitting to the validation set).
> >
> > [1] Systematic generalization with group-invariant predictions
> >
> >
> >
> >
> >
> >
> >
> >
> > **3. Capacity-Disentanglement Tradeoff.**
> >
> > **Answer**: We already discussed this in Appendix F in previous version. Here we provide more details and discussion about this problem.
> >
> > In our experiment, we fix the model dimension $d$ and tune the number of module $K$, so that each module's dimension is $\frac{d}{K}$. This ensure the overall model parameter is the same.
> > Noted that with a fixed number of dimension $d$, with larger $K$, the model capacity is reduced. The widely adopted disentangled representation learning methods like BetaVAE mostly assume that every dimension is mutually independent ($K=d$ in our case), which restricts the model capacity to extreme case. And the results in Table 3 also show that our current disentangled approach performs empirically better than BetaVAE. One potential reason is that we choose allow a middle K is a trade-off between model disentanglement and capacity, in which only the dimension across the block is de-correlated, why the ones within block could still correlated, so as to maintain model expressiveness. (as shown in Figure 7)
> >
> > Note that the optimal choice of $K$ should ideally should be proportional to number of true generative factors that are related to downstream tasks. Therefore, for dataset with a large amount of tasks, we should choose larger number of $K$, and also consider increasing the total number of dimension $d$ to increase the model capacity while maintaining disentanglement. In our paper for large dataset such as Taskonnomy we didn't do further tuning due to limited resources, so the performance could be potentially further improved, which we leave for future exploration.

---

> > > ### Comment · Reviewer_nWqE · 2022-08-06
> > > **response to authors**
> > >
> > > Thank you for your response. The authors provide experimental evidence that addressed my concerns, I am willing to increase the score to 7.

---

> > > > ### Author Response · Authors · 2022-08-09
> > > > **Thanks for the reviews and comments.**
> > > >
> > > > We really appreciate your acknowledgment of our work and your intuitive comments that helped us to improve our paper.

---

### Official Review · Reviewer_ZKY5 · 2022-07-10

**Rating:** 6
**Confidence:** 3
**Soundness:** 3 good
**Presentation:** 4 excellent
**Contribution:** 3 good

**Summary:**

The submission points out the risk of including non-causal features for one task when learning representations in the multi-task settings, accompanied by both a theoretical proof and an empirical analysis.
It then proposes a graph-disentanglement-based method, which is inspired by the disentangled causal mechanism and the invariant risk minimization, to address the problem.

**Questions:**

Please see the Weaknesses listed above.

**Limitations:**

The paper has provided a clear statement on its limitations after the conclusion.

**Strengths And Weaknesses:**

Pros:

- It points out a novel yet important research problem -- the risk of including non-causal features for one task when learning representations in the multi-task settings, as shown in Figure 2.

- It also gives theoretical proof as well as an empirical experiment on synthetic data to show that the problem does exist.

- The proposed causal-graph-based method outperforms previous multi-task methods such as PCGrad and GradVec as well as single-task causal methods such as IRM on multiple multitask datasets that contain unknown distribution shifts.

Cons:

- The disentanglement regularization shown in Eq. (1) assumes **linear correlation**, which will fail miserably with a deep feature extractor. It is highly possible that, with the regularization enforced, the multiple neural modules can still possess the same information even if they are linearly decorrelated (in other words, they can be **non-linearly correlated**).

- As pointed out in Line 225, ``if the complexity of a task-predictor is much larger than the number of environments, it could learn an over-fitted solution that makes gradient zero but does not achieve invariance.'' It seems like the proposed alternative (Eq 8 and Eq 9) can still face the same issue. Have I misunderstood anything?

- It contains a vast number of hyperparameters. It seems possible that the performance gain may come from hyperparameter tuning rather than from removing the spurious correlation. Is there any analysis of hyperparameter sensitivity? Does the proposed method underperform the baselines if the hyperparameters are not appropriately set?

---

> ### Author Response · Authors · 2022-08-02
> **Response to Reviewer ZKY5 [1/3]**
>
> Thanks to the reviewer for recognizing our studied problem as novel and important, and pointing out several key questions of model design. Below we provide our answers to your questions:
>
> **1. It contains a vast number of hyperparameters. It seems possible that the performance gain may come from hyperparameter tuning rather than from removing the spurious correlation. Is there any analysis of hyperparameter sensitivity?**
>
> **Answer**: We added the detailed procedure of hyper-parameter selection and results in **Appendix H**. Here we highlight the procedure and results of hyper-parameter tuning.
>
> Before discussing hyper-parameter selection, let me explain our baseline setup and experiment setting again. We have a single validation set, potentially OOD to the training set. Our method only uses the training set to calculate the loss to update both encoder and per-task predictors. We then use the hold-out validation set to calculate the loss w.r.t graph weights A, and use it to calculate invariance regularization (G-IRM). This avoids overfitting the validation set.
>
>
> We mainly split the hyper-parameters into two sets:
> 1) General hyper-parameter related to all baselines (including ours), including number of hidden states, optimizer, learning rate, number of modules ($K$)
> 2) Our MT-CRL specific hyper-parameter, including weights for disentanglement ($\lambda_{decor}$), sparsity ($\lambda_{sps}$), balance ($\lambda_{bal}$), and invariance ($\lambda_{G-IRM}$)
>
>
> For both sets of hyper-parameters, we tune on the same OOD validation set we used for our method.
> All hyper-parameters except $K$ are standard hyper-parameters for the MTL model. For datasets CityScape, NYUv2, and Taskonomy, we directly use the reported hyper-parameter and dataset setting in previous papers [1,2], in order to achieve a fair comparison. For MultiMNIST and MovieLens, we conduct a grid search for basic parameters, including the number of layers, number of hidden dimensions, optimizer, and learning rate, on the Vanilla MMoE MTL model without regularization. For the number of module ($K$), we only tune on MultiMNIST dataset, and find the optimal value $K=8$ and use for all other datasets (please refer to **Appendix F** for more details). After we determine these general hyper-parameters, we fix them and use them for all different MTL methods. This makes the comparison fair and ensures our performance improvement is not due to extensive hyper-parameter tuning of our method.
>
>
>
> Next, we tune the MT-CRL-specific hyper-parameters on the validation set. Again, we utilize the same OOV validation set to conduct hyper-parameter tuning, as this validation set is not used to calculate training loss and be used to update encoder and per-task predictors. Only the Task-to-Module graph weights receives invariance signal, which only take a tiny portion of the whole model. (If we observe datasets from multiple environments, it would be a better choice to hold out a few environments to serve as the validation set.)
>
>
> Note that four regularization weight terms exist to be tuned, which has a large search space, especially for large datasets. Therefore, we only use Multi-MNIST, the smallest dataset in all our testbeds, to conduct hyper-parameter tuning for the ML-CRL-specific hyper-parameters with grid-search. This is definitely not the best choice, and tuning for each dataset could potentially improve our performance further, but that only makes our improvement higher while not changing the main conclusion of this paper. Specifically, we choose the several ranges for the four regularization weights:
>
> -  Sparse weight  ($\lambda_{sps}$): [$0.0, 0.1, 0.2, 0.5, 1.0, 2.0$]
> - Balancing weight  ($\lambda_{bal}$): [$0.0, 0.2, 0.5, 1.0, 2.0, 5.0$]
> -  Disentanglement weight  ($\lambda_{decor}$): [$0.0, 1.0, 2.0, 5.0, 10.0, 20.0$]
> -  Invariance weight  ($\lambda_{G-IRM^{Var}}$): [$0.0, 5.0, 10.0, 20.0, 50.0, 100.0$]
>
> These ranges are selected by running a few samples to determine that the maximum value that should be within this range, and we keep each selection list to be a length of 6.
> We report the boxplot of detailed results for each regularization weights in Figure 17-20 in the appendix. As is illustrated, for all the regularization, using it is better than not using it ($\lambda=0$), showing their advantage in making our MT-CRL pipeline works. We then select the optimal hyperparameter that achieves the highest validation accuracy, which is $\lambda_{sps}=0.2,\lambda_{bal}=5.0, \lambda_{decor}=20, \lambda_{G-IRM}=5.0$. Further tuning them on a dedicated dataset should potentially bring better performance, but we did not do it to avoid the possibility that our method's performance improvement is brought by extensive hyper-parameter tuning.
>
> [1] The effectiveness of memory replay in large-scale continual learning
>
> [2] Gradient Surgery for Multi-Task Learning

---

> > ### Author Response · Authors · 2022-08-02
> > **Response to Reviewer ZKY5 [2/3]**
> >
> > **2. Does the proposed method underperform the baselines if the hyperparameters are not appropriately set?**
> >
> > **Answer**: From the curve and also the definition of these regularization, we know that for all other terms except sparsity regularization $\lambda_{sps}$, increasing the regularization weight and strictly force model to be balance, de-correlated or invariant doesn't harm too much to the model training (trend didn't go down even with relatively large weight). The only exception is the sparsity regularization. With high $\lambda_{sps}$ implemented as $L_1$ loss over adjacency weights will force all to be zero, which is very harmful to model training, which is why by default, we choose the value as $\lambda_{sps}=0.2$.
> >
> >
> > To give a simple example of whether our model is sensitive to an inappropriate setting of hyper-parameter, we run experiment on MultiMNIST and MovieLens, with the following randomly chosen hyper-parameter setting: $\lambda_{sps}=2.0,\lambda_{bal}=1.0, \lambda_{decor}=2.0, \lambda_{G-IRM^{Var}}=100.0$. The results compared with the original results are as shown below:
> >
> > |   Methods   | Multi-MNIST Accuracy | MovieLens MSE |
> > |:--------:|:------:|:------:|
> > |MT-CRL with $\mathcal{L}_{G\text{-}IRM}^{Var}$ and **default** hyper-parameter|0.915 $\pm$ 0.018 |  0.884 $\pm$ 0.006|
> > |MT-CRL with $\mathcal{L}_{G\text{-}IRM}^{Var}$ and **randomly chosen** hyper-parameter|0.904 $\pm$ 0.021 |  0.887 $\pm$ 0.006|
> > |Vanilla MTL baseline|0.846 $\pm$ 0.018 |0.892 $\pm$ 0.005|
> >
> >
> > Note that with a randomly chosen hyper-parameter, the results on the two datasets drop slightly but are still significantly higher than the Vanilla MTL baseline. This is an informal showcase of our method's generality and not very sensitive to hyper-parameter selection. If time permits, we plan to add more combinations of these hyper-parameters and component choices on more datasets to make our ablation study more convincing.
> >
> > **3. The disentanglement regularization shown in Eq. (1) only regularizes  linear correlation instead of non-linear correlation (the true disentanglement.**
> >
> > **Answer**: The reviewer points out our method's critical design choice (and limitation). In previous version, we’ve already discussed it in Appendix C. Here we elaborate this design choice and why in this paper we choose to only regularize linear correlation.
> >
> > Initially, we plan to regularize non-linear correlation (we tried to minimize estimated mutual information). Minimizing mutual information (MI) requires either knowing the latent distribution (e.g., InfoGAN. We report BetaVAE in Sec 5.1 with similar intuition but performs worse) or over estimated MI (e.g., MINE). We tried adding a discriminator for every module pair and adopted Minmax training to minimize estimated MINE. The result is unstable and no better. Module output's norm is very large, and only the centers are separated rather than disentangled. Specifically, the results on Multi-MNIST are 0.874, while the beta-VAE is 0.896, and we eventually adopted linear correlation regularization is 0.915. We did not report the result in the original paper with adversarial minimizing estimated MINE as it is not a formal method and we decide the linear correlation method that has better empirical results.
> >
> >
> > To explain this observation, we hypothesize that the non-linear disentangled method we tried might restrict the model capacity too much and contain information unrelated to tasks. For example, most existing VAE-based disentangled representation learning methods require storing all latent generative factors that are useful to reconstruct the data. However, it's possible that some factors are not related to any of the tasks. On the other hand, minimizing estimated MINE requires solving a min-max game which is very unstable for classification tasks. On the other hand, our proposed method regularizes the linear correlation between any pair of output dimensions. This method is simple and works for our current setting. Therefore, although adding non-linear disentangled loss is more theoretically appealing, we still only use the linear version in the current paper. We have updated the limitation statement and added "not regularizing non-linear correlation in disentanglement regularization" as our limitation. We leave more exploration on this part as future work (if the reviewer knows some effective methods to disentangle modular output in a supervised setting, please let us know, and we are definitely glad to try it out). We’ve added this point into our limitation statement and leave for future work.

---

> > > ### Author Response · Authors · 2022-08-02
> > > **Response to Reviewer ZKY5 [3/3]**
> > >
> > > **4. Given a limited number of environments, will the gradient become zero by optimizing eq 8 and 9.**
> > >
> > > **Answer**: Firstly, the meaning of eq 8 and 9 is that when we calculate the invariance score, we only calculate the loss over the causal graph weights (A), which only have parameter #module $*$ #task and are not likely to overfit. As reviewer vBSD points out, a similar trick has been adopted in eq 6 in [1].
> > >
> > > In addition, another potential issue is that if the number of training samples (not the number of environments) for each environment is too small, the predictor could easily overfit all training samples and get near-zero gradients. In our experiments, we only consider a training set and a validation set as two environments to avoid this issue. We only take the training set to calculate the loss to update both encoder and per-task predictors. We then use the hold-out validation set to calculate the loss w.r.t graph weights A, and update it via invariance regularization. (This avoids overfitting to the validation set).
> > >
> > > [1] Systematic generalization with group-invariant predictions

---

### Official Review · Reviewer_vBSD · 2022-07-11

**Rating:** 7
**Confidence:** 4
**Soundness:** 3 good
**Presentation:** 3 good
**Contribution:** 3 good

**Summary:**

The submission discusses a way in which multi-task learning can potentially promote the use of features that might generalize poorly in OOD settings. This can happen when labels for different tasks (involving different features) are correlated in the training set but not outside of it.

A method is proposed to improve generalization in such situations by imposing (soft) modularity on the predictive format. Specifically, a (soft) sparse connectivity matrix is learned connecting sets of encodings (modules) to tasks, with training losses to encourage de-correlation among the modules, sparse module-usage, and discourage degenerate solutions such as dead modules. Finally, an invariance penalty inspired from IRMv1 [1] and V-REx [2] is applied on the connectivity matrix to promote the same weighting of modules across different environments, with the assumption that spurious label-label correlations across MTL tasks are likely to be unstable across different environments.

Experiments on a range of synthetic and realistic datasets (with synthetic splitting in all cases to simulate IID-OOD settings) showcase improvements over baselines.


[1] Invariant risk minimization, Arjovsky et al., 2019.

[2] OOD generalization via risk extrapolation, Krueger et al., 2020.

**Questions:**

1. The most major question I have is: how is hyper-parameter selection being performed? The precise method is unclear to me from my reading, but my current understanding is that since training and validation splits correspond to two environments with the invariance penalty applied for these two sets, non-invariance hyper-parameters are tuned on the validation set. If this is the case, how is the invariance penalty hyper-parameter tuned? Is this done at a second stage, with non-invariance hyper-parameters fixed from the previous stage? A related question: for every baseline method, are all hyper-parameter tuning-choices made on the same OOD validation set as for G-IRM?


2. For the modular architecture, was the feature dimension scaled up by K, or divided into K parts from the same dimensionality as in baselines? This could imply different choices for baseline comparisons.


**Limitations:**

The authors have provided a limitation-statement.

**Strengths And Weaknesses:**

The discussion of the possibility of a multi-task learning setup inadvertently promoting the learning of less-stable features due to label correlations among the tasks is original, to my knowledge. The robustness-method is largely derived from existing works for the most part, but there are some elements of interest. For example, fixing f_t in the invariance penalty is an interesting trick (although something similar was done in [3]). The case study with MovieLens seems like a nice example of how such task-label correlations could arise.

I find the overall quality of the paper to be good. There are a number of relatively minor typos/writing issues I’ve detailed below, but I found the presentation reasonably clear on the whole.

Miscellaneous thoughts:

 — In L126, it might make more sense to rephrase “cannot generalize” as “perform relatively less effectively”, since presumably the correlation between the label and the causal factors is much stronger (for example, Table 1 only shows 2-3% drops and not utter failure). This also suggests a natural baseline can often simply be ERM training drawn out over more epochs/iterations, since with longer training, gradient descent can often be expected to unlearn any high-bias but non-stable features (the “noisy correlations” in L218-L220). In reality, for neural network training, other aspects to do with training dynamics crop up which might well lead to more drastic failure, but for the purposes of the theoretical arguments being made here, the outlook is less gloomy.

 — As acknowledged in the submission, invariance across a sampling of environments is not guaranteed to capture causality, and comes with several assumptions. Perhaps it is more appropriate to not imply that works such as these achieve causal discovery or even approximate it, but rather that they simply learn robust features, which are more likely to be stable in OOD settings. Also, since A is a soft-matrix (even with the sparsity penalty), it’s not clear to me if it makes sense to refer to it as a causal “graph”.

 — I’m not sure if label-label correlations are primarily the issue; the real problem seems to be the feature-feature correlations, if I’ve understood the setup correctly (and this boils down to the usual feature-label correlations). The two seem equivalent, but consider introducing label-noise: now the label-label correlation rate can drop, and depending on the strengths of the correlation-rate and the label-noise, the label-label correlation can drop to negligible rates while the feature-feature correlation rate stays where it was. I understand the point in the submission that the underlying reason behind the correlation is that the MTL setup promotes the correlation, but I was wondering if a more direct way to view this problem would be that the features corresponding to the two tasks are correlated, rather than the labels. Just a suggestion for the authors to consider.


[3] Systematic generalization with group-invariant predictions, Ahmed et al., 2021.

Typos:

 — Fig 2 caption should say F_b instead of F_a both times

 — Missing period at end of title for Section 2.1

 — L133: “exploit” —> “exploits”

 — L143: “The” —> “the”

 — L170: “extracts” —> “extract”

 — L177: “prediction” —> “predictions”

 — L183: “disentangle” —> “disentangled”?

 — L186: Missing period at end of sentence.

 — L186-L187: Cheung 2015 and Cogswell 2016 are described as involving “generative disentangled representation works”, but I believe both of these works deal with disentangling representations in classification settings, not generative.

 — L211: the per-task predictor was defined as f_t, not g.

 — L215: “exist” —> “exhibit”?

 (Few others, running spell-check should fix)

==================================
Post-rebuttal: I've read the rebuttal, and it answers my questions adequately on the whole. I'm raising my score to 7.

---

> ### Author Response · Authors · 2022-08-02
> **Response to Reviewer vBSD [1/4]**
>
> Thanks to the reviewer for the in-depth question about the setting of our paper. We’ve modified the paper and provided some explanation here and hope it could address some of the concerns:
>
> **1. How is hyper-parameter selection being performed for our method and baselines?**
>
> **Answer**: We added the detailed procedure of hyper-parameter selection and results in **Appendix H**. Here we highlight the procedure and results of hyper-parameter tuning.
>
> Before discussing hyper-parameter selection, let me explain our baseline setup and experiment setting again. We have a single validation set, potentially OOD to the training set. Our method only uses the training set to calculate the loss to update both encoder and per-task predictors. We then use the hold-out validation set to calculate the loss w.r.t graph weights A, and use it to calculate invariance regularization (G-IRM). This avoids overfitting the validation set.
>
>
> We mainly split the hyper-parameters into two sets:
> 1) General hyper-parameter related to all baselines (including ours), including number of hidden states, optimizer, learning rate, number of modules ($K$)
> 2) Our MT-CRL specific hyper-parameter, including weights for disentanglement ($\lambda_{decor}$), sparsity ($\lambda_{sps}$), balance ($\lambda_{bal}$), and invariance ($\lambda_{G-IRM}$)
>
>
> For both sets of hyper-parameters, we tune on the same OOD validation set we used for our method.
> All hyper-parameters except $K$ are standard hyper-parameters for the MTL model. For datasets CityScape, NYUv2, and Taskonomy, we directly use the reported hyper-parameter and dataset setting in previous papers [1,2], in order to achieve a fair comparison. For MultiMNIST and MovieLens, we conduct a grid search for basic parameters, including the number of layers, number of hidden dimensions, optimizer, and learning rate, on the Vanilla MMoE MTL model without regularization. For the number of module ($K$), we only tune on MultiMNIST dataset, and find the optimal value $K=8$ and use for all other datasets (please refer to **Appendix F** for more details). After we determine these general hyper-parameters, we fix them and use them for all different MTL methods. This makes the comparison fair and ensures our performance improvement is not due to extensive hyper-parameter tuning of our method.
>
>
>
> Next, we tune the MT-CRL-specific hyper-parameters on the validation set. Again, we utilize the same OOV validation set to conduct hyper-parameter tuning, as this validation set is not used to calculate training loss and be used to update encoder and per-task predictors. Only the Task-to-Module graph weights receives invariance signal, which only take a tiny portion of the whole model. (If we observe datasets from multiple environments, it would be a better choice to hold out a few environments to serve as the validation set.)
>
>
> Note that four regularization weight terms exist to be tuned, which has a large search space, especially for large datasets. Therefore, we only use Multi-MNIST, the smallest dataset in all our testbeds, to conduct hyper-parameter tuning for the ML-CRL-specific hyper-parameters with grid-search. This is definitely not the best choice, and tuning for each dataset could potentially improve our performance further, but that only makes our improvement higher while not changing the main conclusion of this paper. Specifically, we choose the several ranges for the four regularization weights:
>
> -  Sparse weight  ($\lambda_{sps}$): [$0.0, 0.1, 0.2, 0.5, 1.0, 2.0$]
> - Balancing weight  ($\lambda_{bal}$): [$0.0, 0.2, 0.5, 1.0, 2.0, 5.0$]
> -  Disentanglement weight  ($\lambda_{decor}$): [$0.0, 1.0, 2.0, 5.0, 10.0, 20.0$]
> -  Invariance weight  ($\lambda_{G-IRM^{Var}}$): [$0.0, 5.0, 10.0, 20.0, 50.0, 100.0$]
>
> These ranges are selected by running a few samples to determine that the maximum value that should be within this range, and we keep each selection list to be a length of 6.
> We report the boxplot of detailed results for each regularization weights in Figure 17-20 in the appendix. As is illustrated, for all the regularization, using it is better than not using it ($\lambda=0$), showing their advantage in making our MT-CRL pipeline works. We then select the optimal hyperparameter that achieves the highest validation accuracy, which is $\lambda_{sps}=0.2,\lambda_{bal}=5.0, \lambda_{decor}=20, \lambda_{G-IRM}=5.0$. Further tuning them on a dedicated dataset should potentially bring better performance, but we did not do it to avoid the possibility that our method's performance improvement is brought by extensive hyper-parameter tuning.
>
> [1] The effectiveness of memory replay in large-scale continual learning
>
> [2] Gradient Surgery for Multi-Task Learning

---

> > ### Author Response · Authors · 2022-08-02
> > **Response to Reviewer vBSD [2/4]**
> >
> > (continue the first question about hyper-parameter tuning)
> >
> > We also add a simple sensitivity analysis by choosing a randomly chosen hyper-parameter combination $\lambda_{sps}=2.0,\lambda_{bal}=1.0, \lambda_{decor}=2.0, \lambda_{G-IRM^{Var}}=100.0$. The results are as below:
> >
> > |   Methods   | Multi-MNIST Accuracy | MovieLens MSE |
> > |:--------:|:------:|:------:|
> > |MT-CRL with $\mathcal{L}_{G\text{-}IRM}^{Var}$ and **default** hyper-parameter|0.915 $\pm$ 0.018 |  0.884 $\pm$ 0.006|
> > |MT-CRL with $\mathcal{L}_{G\text{-}IRM}^{Var}$ and **randomly chosen** hyper-parameter|0.904 $\pm$ 0.021 |  0.887 $\pm$ 0.006|
> > |Vanilla MTL baseline|0.846 $\pm$ 0.018 |0.892 $\pm$ 0.005|
> >
> >
> > Randomly chosen setting are lower than using default setting, but still significantly higher than the vanilla MTL model. This is an informal showcase of our method's generality and not very sensitive to hyper-parameter selection. If time permits, we plan to add more combinations of these hyper-parameters and component choices on more datasets to make our ablation study more convincing.
> >
> > **2. Was the feature dimension scaled up or divided by K.**
> >
> > **Answer**: It is divided by K. (We have discussed the details of selecting K in **Appendix F**). The reason for choosing "divide" is to ensure the hidden dimension is always the same across different methods and baselines. By increasing K, the more rigorous regularization will limit model capacity. In this way, all the methods compared have the same number of parameters.
> >
> > **3. In L126, it might make more sense to rephrase “cannot generalize” as “perform relatively less effectively”, and a gradient descent baseline can be expected to unlearn any high-based but non-stable features.**
> >
> >
> > **Answer**: We agree that “cannot” is too absolute as non-causal factors shall influence the prediction but not make it fail, so we change the wording as suggested. On the other hand, we disagree that ERM trained longer is a solution for such an OOD problem, but merely a baseline. As we discussed in the paper, there exist two kinds of spurious features:
> >
> > - The first kind of spurious feature is caused by confounders (either feature-label confounders in traditional single-task learning setup or label-label ones discussed in our paper), which is a systematic error. In this case, some research shows that with noise on causal features, the model will inevitably utilize spurious ones [1]. Some show that even in a noiseless and overparametrized setting (so that it is easy to achieve 0 training error) with weight decay, the model will still assign non-zero weights to spurious features to fit data with smaller weight norm. (eq 7 in [2]). Similarly, we also prove in Appendix A.1 that with label-label confounders, even the Bayes optimal model with unlimited data points will inevitably use spurious features. In this case, we cannot simply use gradient descent to unlearn spurious features (without explicit regularization). Results in Table 1 could also support this. With the confounder (we manually construct for synthetic analysis), even though the training accuracy is already very high, MTL generalization is still much worse, and the ratio of spurious feature usage is higher (in this experiment, we train the model for sufficient epochs, so the model indeed converge)
> >
> > - The second kind of spurious feature is only caused by noise. In this situation, the spurious feature could be removed with a sufficient number of training samples and an expressive model and optimization. Training a neural network with limited samples is much more complicated. Some theorems have shown that a neural network trained with gradient descent has a small generalization error bound [3] with proper initialization, but still it’s not guaranteed that all noise could be removed and achieve 0 generalization error. Our method in this case, could potentially utilize a validation set as guidance to reduce further the usage of noise (still, not guaranteed but just empirically).
> >
> > To sum up, with both cases of spurious features, we do not think vanilla ERM without explicit regularization is enough to remove them as they are “low-variance” and achieve good OOD generalization. In our experiment, we also have ERM baselines, and all these baselines are trained with sufficient epochs until the validation error does not decrease. Please correct me if the reviewer knows some recent study that shows such properties of neural network training.
> >
> >
> > [1] Understanding the Failure Modes of Out-of-Distribution Generalization
> >
> > [2] Removing spurious features can hurt accuracy and affect groups disproportionately
> >
> > [3] Generalization Error Bounds of Gradient Descent for Learning Over-Parameterized Deep ReLU Networks

---

> > > ### Author Response · Authors · 2022-08-02
> > > **Response to Reviewer vBSD [3/4]**
> > >
> > > **4. A is a soft matrix and whether it makes sense to be called causal “graph”.**
> > >
> > > **Answer**: Firstly, our terminology of the causal graph is based on previous literature on causal structure learning [1,2,3]. (For example, check out eq 8 in [1]). We agree that this terminology might bring some confusion, so we modified the wording to define A as a weighted adjacency matrix instead of calling it a “graph”. Secondly, we want to point out that after adding the L0 sparse regularization, the final learned A tends to be sparse. Please refer to Figures 11 and 13 in Appendix. For those weights the model recognizes as less informative, it converges to be a very low score close to 0. We added a simple experiment on Multi-MNIST by adopting a threshold of 0.1 for the adjacency weight (all weights lower than 0.1 will be shrinkage to be 0), so the adjacency matrix is indeed discrete. The average performance of such a discrete version is 0.917, even higher than the soft version (0.915 shown in Table 6). This implies that such discretization of graph modeling indeed helps OOD generalization. One potential extension is to utilize differentiably sparse gate modeling [4] in our framework, which we leave for future works.
> > >
> > > **5. Label-label correlation compared with feature-feature correlation.**
> > >
> > > **Answer**: Thanks to the reviewer for pointing out this fundamental question. Though there indeed exist a few works studying feature-feature correlation in single-task learning [5], we find it very difficult to define and model in multi-task settings. The key differences are:
> > > The causal features/factors for different tasks could overlap in multi-task learning—for example, the $F_i$ in Figure 2 of our paper. Thus, we must consider which part of the feature is causally related to each task and their correlation.
> > >
> > > If we consider a feature causally related to both tasks, we need to model task-conditioned distribution, e.g., the correlation between P($F_i$ | $Y_a$) and P($F_i$ | $Y_b$). Compared with a task label, usually a single digit, such conditional distribution is much harder to model, especially for high-dimensional features.
> > >
> > > Whether label or feature is more fundamental depends on the direction of the causal mechanism. In the literature on causal models, there exists a causal (X -> Y) and anti-causal (Y -> X) modeling framework [6]. In our paper, we follow the second setting, in which the task labels could be regarded as high-level concept symbols (e.g., color, shape, object, etc.) to describe data. These variables generate the feature X via P(X|Y). In this framework, considering label-label confounders is more fundamental to feature-feature, as the latter is caused by the former. (assuming we did not consider other confounders that could influence feature generation).
> > >
> > > In real-world modeling, the factors we consider should be latent representation instead of discrete features (as the raw input such as pixel usually is less informative than intermediate learned representation). In that case, it is hard to model the correlation of learned representation output by a continuously updated encoder. Therefore, in our model, we simply adopt disentangled regularization to get multiple feature views. We also focus on modeling the task-to-module correlation under different environments (with potentially label-label confounder changes). Therefore, the modeling matches with a real model design better.
> > >
> > > We agree that considering feature correlation is a very interesting research topic, especially for the tasks where the input feature is disentangled and contains rich semantic meaning (e.g., over graph). However, given that this paper focuses on studying spurious correlation in multi-task learning, we think label-label confounder is a more natural choice.
> > >
> > >
> > >
> > >
> > > [1] DAGs with NO-TEARS: Continuous Optimization for Structure Learning
> > >
> > > [2] Masked Gradient-Based Causal Structure Learning
> > >
> > > [3] GRADIENT-BASED NEURAL DAG LEARNING
> > >
> > > [4] DSelect-k: Differentiable Selection in the Mixture of Experts with Applications to Multi-Task Learning
> > >
> > > [5] Removing spurious features can hurt accuracy and affect groups disproportionately
> > >
> > > [6] On Causal and Anticausal Learning

---

> > > > ### Author Response · Authors · 2022-08-02
> > > > **Response to Reviewer vBSD [4/4]**
> > > >
> > > > **6. Invariance is not guaranteed to capture causality.**
> > > >
> > > > **Answer**: Many existing works studying “invariance” property (including IRM and V-Rex) point out the connection between invariance and causality (see section 4 in IRM, section 3.2 in Rex, as well as ICP paper). However, their proofs have many assumptions, such as only based on linear structural causal models, a sufficient number of environments, etc. To this extent, we did not expect our framework based on these works, nor the deep neural Multi-Task model can recover ground-truth causal mechanisms (and we did not claim this in the paper). Instead, we expect the used architecture motivated by existing causal mechanism modeling could improve robustness (as suggested by the reviewer). We modified the wording in the approach section to avoid misunderstanding.
> > > >
> > > >
> > > >
> > > >
> > > > **7. One related work "Systematic generalization with group-invariant predictions"**
> > > >
> > > > **Answer**: Thanks for pointing out this reference. We have added this work in the approach section when we introduce the Graph-Invariant Risk Minimization. Indeed the intuition is similar, but this work focuses on learning dataset split while we are learning task-to-module graphs. We've added some discussion with this work when we introduce our G-IRM method.

---

### Official Review · Reviewer_kRci · 2022-07-14

**Rating:** 7
**Confidence:** 4
**Soundness:** 3 good
**Presentation:** 3 good
**Contribution:** 3 good

**Summary:**

This paper studies the problem of the lack of the generalizability of multi-task learning due to the spurious (no causal relationships) correlations between tasks. Such spurious correlations may wrongly be used during the learning process leading to bad generalization. Furthermore, similar spurious correlations on the label side are also explored. In the setting of label-label confounders, this paper shows that multi-task learning is prone to exploiting the undesired non-causal knowledge from other tasks. The argued problem of spurious correlations is tackled by using the proposed multi-task causal representation (MT-CRL) framework, MT-CRL represents multitask knowledge in a disentangled fashion and exploits the MTL-specific invariant regularization to learn the causality. The paper conducts experiments on multiple benchmark papers to demonstrate the effectiveness of the proposed method.

**Questions:**

Most of my questions are answered in the supplementary material. I am wondering if the task similarly graph of Figure 9 (in supplementary material) for Taskonomy datasets looks similar to the Taskonomy datasets or not?


**Limitations:**

There is no obvious severe limitation of this paper besides what the authors have already mentioned in the paper.
I can say nothing regarding the negative social impact of this paper from my reading.


**Strengths And Weaknesses:**

Strength:

* The problem studied in the paper, although not very original, is meaningful in the context of multi-task learning.

* The proposed method, especially causal learning by graph invariant regularization is interesting. The practical consideration on optimizing the intractable bi-level optimization using the graph-invariant risk minimization is a valuable technical contribution.

* The paper is well written and easy to follow due to its structure, examples, and interesting discussion.

* The experimental results validate the effectiveness of the proposed method over the established baseline. Experiments on five benchmark datasets make the experimental section solid.

* Supplementary material provides additional analysis, discussion, and results that further support the contributions.

Weakness:

* This paper has no serious weakness except the missing evaluation in the case of known confounder changes. However, this weakness has clearly been discussed in the limitations.

* The contribution of the term L_G_{IRM}^var is rather marginal in Table 2.

* Only the relative improvements (Table 2) and losses (Table 8, supplementary material) are reported on Taskonomy dataset. The performance evaluation on each task and the improvements would be interesting. Furthermore, only a subset of the available tasks is used.

* Ablation results reported in Table 3 appear to be incomplete. Only a subset of cases is reported.

* Minor: equations are not punctuated. Typo in L201.

---

> ### Author Response · Authors · 2022-08-02
> **Response to Reviewer kRci**
>
> We thank the reviewer for constructive comments. As the reviewer points out, many of the concerns have already been discussed in the appendix of our previous version. Here we highlight them out and add more explanations. Hope it could help clarify and solve the reviewer's concern.
>
> **1. Task Similarity Graph for Tasknomy.**
>
> **Answer**: Thanks for pointing this out. We have added the Task-to-Module graph as well as the induced similarity matrix in **Appendix G**. As is shown in the figure, some similar task like 2D keypoint Estimation and 3D keypoint are liked together, and also the hard task like semantic segmentation receives information from curvature estimation, surface normal estimation and occlusion edge detection. These findings fit the observation of original Taskonomy paper.
>
> **2. More Ablation Studies.**
>
> **Answer**: From our understanding, the most important model design of our paper is: 1) disentanglement; 2) graph regularization; 3) invariance regularization. Instead of enumerating the combination of these three components, we mainly follow the leave-one ablation, i.e., remove each of them and keep all others the same, which is the last block of Table 2 and Table 3. For other important hyper-parameter selection and sensitivity analysis, please check our added **Appendix H**. If time permits, we plan to add more combinations of these hyper-parameters and component choices on more datasets to make our ablation study more convincing.
>
> **3. Missing evaluation in the case of known confounder changes.**
>
> **Answer**: As discussed in the limitation statement, we currently only have two synthetic datasets with known confounder changes (in Sec. 2). We plan to construct a more comprehensive benchmark (e.g., based on existing disentangled representation learning datasets like dSprites, Shapes3D, etc) in future work.
>
> **4. The contribution of $L_{G-IRM}^{Var}$ is rather marginal.**
>
> **Answer**: I assume the reviewer is talking about the improvement of $L_{G-IRM}^{Var}$ compared with $L_{G-IRM}^{Norm}$, as the improvement of our MT-CRL compared with other baselines is significant. (Please correct me if I misunderstand your point.)
>
> As we discussed in the approach section, the main problem of $L_{G-IRM}^{Norm}$ is that it will influence optimization during the initial stage when the model is not trained well and shall have a very large gradient norm.
>
> Among the five datasets we use, the most complicated and large-scale is Taskonomy, which contains eight very different tasks. On this hardest dataset, $L_{G-IRM}^{Var}$ outperforms $L_{G-IRM}^{Norm}$ by 0.6%. This shows that regularizing gradient variance across environments instead of the norm could help deal with complicated MTL datasets with many different tasks.
>
>
> **5. The performance evaluation on each task.**
>
> **Answer**: In Table 6-10 in the Appendix of the original paper submission, we show the detailed performance on each task of our method compared to each baseline.
>
>
>
> **6. Only a subset of the available tasks is used.**
>
> **Answer**: For all the other datasets except Taskonomy, we use all available subtasks to evaluate our method. For Tasknomy, we follow the codebase and setting of [1], as the eight chosen tasks are the most representative in Taxonomy. For a detailed experimental setting on real-world datasets, you can find at Appendix D.2
>
> [1] Balaji, Y., Farajtabar, M., Yin, D., Mott, A., and Li, A. The effectiveness of memory replay in large-scale continual learning

---

### Author Response · Authors · 2022-08-02
**General Response to all Reviewers**

Sincerely thanks all the reviewers for their constructive comments and suggestions for this paper. Here we highlight the major updates of our revised paper (highlighted as red in the paper):

1. Add **detailed hyper-parameter tuning procedure**, results and a sensitivity analysis at **Appendix H**.
2. Add a **case study on Taskonomy** (Task-to-Module Graph and induced Task similarity graph) at **Appendix G**.
3. Add more discussion of capacity-disentanglement trade-off in **Appendix F** and disentanglement in **Appendix I**.

As acknowledged by the reviewers, the primal focus of this paper is to point out **unique challenges and differences of spurious correlation problems in multi-task learning from existing research in single-task learning**. We believe this spurious correlation problem is critical to alleviating negative transfer for multi-task learning, and a good direction for future research to further analyze it and propose better model to solve this problem.

We hope the added description could clarify our paper and solve reviewers’ concern (especially the hyper-parameter tuning part), and glad to have more discussion on other interesting questions and comments.

---

### Meta-Review · Area_Chair_Di8e · 2022-08-27

**Recommendation:** Accept
**Confidence:** Certain

**Metareview:**

This work studies how to alleviate potential spurious correlation in multi-task learning, and proposes a Multi-Task Causal Representation Learning (MT-CRL) framework, which aims to represent multi-task knowledge via disentangled neural modules, and learns which module is causally related to each task via MTL-specific invariant regularization

The proposed method sounds reasonable, the results are solid, and the paper is well written. Overall it is a good work.


**Award:**

No

---

### Decision · Program_Chairs · 2022-09-14

Accept